



# Evaluation of the University of Victoria Earth System Climate Model version 2.10 (UVic ESCM 2.10)

Nadine Mengis[1,2], David P. Keller[1], Andrew MacDougall[3], Michael Eby[4], Nesha Wright[2], Katrin J. Meissner[5], Andreas Oschlies[1], Andreas Schmittner[6], H. Damon Matthews[7], Kirsten Zickfeld[2]

[1]Biogeochemical Modelling Department, GEOMAR – Helmholtz Centre for Ocean Research, Kiel, Germany
[2]Geography Department, Simon Fraser University, Burnaby, BC, Canada
[3]Climate and Environment, St. Francis Xavier University, Antigonish, NS, Canada
[4]School of Earth and Ocean Sciences, University of Victoria, Victoria, BC, Canada
[5]Climate Change Research Centre, The University of New South Wales, Sydney, New South Wales, Australia and The Australian Research Council Centre of Excellence for Climate Extremes, Sydney, New South Wales, Australia
[6]College of Earth, Ocean, and Atmospheric Sciences, Oregon State University, Corvallis, OR, USA
[7]Concordia University, Montréal, Canada

*Correspondence to*: Nadine Mengis (nmengis@geomar.de)





**Abstract.** The University of Victoria Earth system climate model of intermediate complexity has been a useful tool in recent assessments of long-term climate changes, including both paleo-climate modelling and uncertainty assessments of future warming. Since the last official release of the UVic ESCM 2.9, and the two official updates during the last decade, considerable model development has taken place among multiple research groups. The new version 2.10 of the University of Victoria Earth System Climate Model (UVic ESCM) presented here, and to be used in the 6[th] phase of the coupled model intercomparison project (CMIP6), combines and brings together multiple model developments and new components that have taken place since the last official release of the model. To set the foundation of its use, we describe here the UVic ESCM 2.10 and evaluate results from transient historical simulations against observational data. We find that the UVic ESCM 2.10 is capable of reproducing well changes in historical temperature and carbon fluxes, as well as the spatial distribution of many ocean tracers, including temperature, salinity, phosphate and nitrate. This is connected to a good representation of ocean physical properties. For the moment, there remain biases in ocean oxygen, which will be addressed in the next updates to the model.

## 1. Introduction

The University of Victoria Earth System Climate Model (UVic ESCM) of intermediate complexity has been a useful tool in recent assessments of long term climate changes including paleo-climate modelling (e.g. Alexander et al., 2015; Bagniewski et al., 2017; Handiani et al., 2012; Meissner et al., 2003; Menviel et al., 2014), carbon cycle dynamics (e.g. Matthews et al., 2009b; Matthews and Caldeira, 2008; Montenegro et al., 2007; Schmittner et al., 2008; Tokarska and Zickfeld, 2015; Zickfeld et al., 2009, 2011, 2016) and climate change uncertainty assessments (e.g. Ehlert et al., 2018; Leduc et al., 2015; MacDougall et al., 2017; MacDougall and Friedlingstein, 2015; Matthews et al., 2009a; Mengis et al., 2018, 2019; Rennermalm et al., 2006; Taucher and Oschlies, 2011; Zickfeld et al., 2015). The UVic ESCM has been instrumental in establishing the irreversibility of $CO_2$-induced climate change after cessation of $CO_2$ emissions (Matthews et al., 2008; Eby et al., 2009) and the proportional relationship between global warming and cumulative $CO_2$ emissions (Matthews et al., 2009; Zickeld et al., 2009). Since the last official release of the UVic ESCM 2.9, and the two official updates during the last decade (Eby et al., 2009; Zickfeld et al., 2011), there are representations for a new marine ecosystem model (Keller et al., 2012), higher vertically resolved soil dynamics (Avis et al., 2011), and permafrost carbon (MacDougall et al., 2012; MacDougall and Knutti, 2016).

The marine ecosystems and biological processes play an important, but often less understood, role in global biogeochemical cycles. They affect the climate primarily through the "carbonate" and "soft tissue" pumps (i.e., the "biological" pump) (Longhurst and Harrison, 1989; Volk and Hoffert, 1985). The biological pump has been estimated to export between 5 and 20 Gt C yr$^{-1}$ out of the surface layer (Henson et al., 2011; Honjo et al., 2008; Laws et al., 2000). However, as indicated by the large range of estimates there is great uncertainty in our understanding of the magnitude of carbon export (Henson et al., 2011), its sensitivity to environmental change (Löptien and Dietze, 2019) and thus its effect on the Earth's climate. Above that, marine ecosystems also play a large role in the cycling of nitrogen, phosphorus, and oxygen. In surface waters nitrogen





and phosphorus constitute major nutrients that are consumed by, and drive, primary production (PP) and thus are linked back to the carbon cycle.

In the recent special report on limiting global warming to 1.5 °C from the Intergovernmental Panel on Climate Change one of the key uncertainties for the assessment of the remaining global carbon budget was the impact from unrepresented Earth system feedbacks. On the decadal to centennial timescales this specifically refers to the permafrost carbon feedback (Lowe and Bernie, 2018). Quantifying the strength and timing of this permafrost carbon cycle feedback to climate change has been a goal of Earth system modelling in recent years (e.g., Burke et al., 2012; Koven et al., 2011, 2013; MacDougall et al., 2012; Schaefer et al., 2011; Schneider Von Deimling et al., 2012; Zhuang et al., 2006).

For version 2.10 of the University of Victoria Earth system climate model, we combined version 2.9 with the new marine ecosystem model component as published in Keller et al. (2012), and the soil dynamics and permafrost carbon component as published by Avis et al. (2011) and MacDougall & Knutti (2016). For the 6th Phase of the Coupled Model Intercomparison Project (CMIP6) simulations, the merge of these two components will allow a more comprehensive representation of the carbon cycle in the UVic ESCM, while incorporating the model developments that have been taken place in the context of the UVic ESCM. In addition to the structural changes, we also changed the spin-up protocol to follow CMIP6 protocols and applied the newly available CMIP6 forcing.

The objective of the new model development is to have a more realistic representation of carbon and heat fluxes in the UVic ESCM 2.10 in agreement with the available observational data and with current process understanding to be used within the context of the next round of model intercomparison projects for models of intermediate complexity. To set the foundation of its use, we will in the following describe the UVic ESCM 2.10 (Sect. 2.1.), the newly formatted historical CMIP6 forcing that has been and will be used (Sect. 2.2.), explicitly describe changes that have been implemented in the UVic ESCM with respect to the previously published versions (Sect. 2.3.), and then evaluate results from transient historical simulations against observational data (Sect. 3.).

## 2. Methods

### 2.1. Description of the University of Victoria Earth system climate model version 2.10

The UVic ESCM is a model of intermediate complexity (Weaver et al., 2001), all model components have a common horizontal resolution of 3.6° longitude and 1.8° latitude and the oceanic component has a vertical resolution of 19 levels, with vertical thickness varying between 50 m near the surface to 500 m in the deep ocean. The Modular Ocean Model Version 2 (MOM2) (Pacanowski, 1995) describes the ocean physics, it is coupled to a thermodynamic-dynamic sea-ice model (Bitz et al., 2001) with elastic visco-plastic rheology (Hunke and Dukowicz, 1997). The atmosphere is represented by a two-dimensional atmospheric energy moisture balance model (Fanning and Weaver, 1996). Wind velocities used to calculate advection of atmospheric heat and moisture as well as the air-sea-ice fluxes of surface momentum, heat and water fluxes, are prescribed as monthly climatological wind fields from NCAR/NCEP reanalysis data (Eby et al., 2013). In transient simulations





wind anomalies, which are determined from surface pressure anomalies with respect to pre-industrial surface air temperature, are added to the prescribed wind fields (Weaver et al., 2001). In addition, the terrestrial component represents vegetation dynamics including five different plant functional types (Meissner et al., 2003). Sediment processes are represented using an oxic-only calcium-carbonate model (Archer, 1996). Terrestrial weathering is diagnosed from the net sediment flux during spin-up and held fixed at the equilibrium pre-industrial value for transient simulations (Meissner et al., 2012).

The new version 2.10 of the University of Victoria Earth System Climate Model (UVic ESCM) presented here combines and brings together multiple model developments and new components that have taken place since the last official release of the model in the CMIP5 context. In the following the novel model components are described in detail.

(i) Marine biogeochemical model

Novel compared to the 2009 version of the model, is the ocean biogeochemistry model as published by Keller et al. (2012). It now includes equations describing phytoplankton light limitation and zooplankton grazing, a more realistic zooplankton growth and grazing model, and formulations for an iron limitation scheme to constrain phytoplankton growth. In this context the ocean's mixing scheme was changed from a Bryan-Lewis profile to a scheme for the computation of tidally induced diapycnal mixing over rough topography (Simmons et al., 2004) (see ocean diffusivity profiles in Figure S3). In addition, the air to sea gas parameterization was updated following the ocean carbon-cycle model intercomparison project updates for these numbers (Wanninkhof, 2014), which impacts the carbon exchange between the atmospheric and marine components. Furthermore, we now apply the stoichiometry from Paulmier et al. (2009) to consistently account for the effects of denitrification and nitrogen fixation on alkalinity and oxygen.

(ii)    Soil model

The terrestrial component has also been updated relative to the latest official release of the UVic ESCM. It now includes a representation of soil freeze–thaw processes resolved in 14 subsurface layers of which the thicknesses exponentially increase with depth: the surface layer having a thickness of 0.1 m and the bottom layer a thickness of 104.4 m, the total thickness of the subsurface layers is 250 m. The top eight layers (to a depth of 10 m) are soil layers; below this are bedrock layers having the thermal characteristics of granitic rock. Moisture undergoes free drainage from the base of the soil layers and the bedrock layers are hydrologically inactive (Avis et al., 2011). In addition, the soil module includes a multi-layer representation of soil carbon (MacDougall et al., 2012). Organic carbon from the litter flux is allocated to soil layers as a decreasing function of depth, and is only added to soil layers with a temperature above 1ºC. If all layers are below this temperature threshold the litter flux is added to the top layer of soil. Soil respiration remains a function of temperature and moisture (Meissner et al., 2003) but is now implemented in each layer individually. Respiration ceases if the soil layer is below 0ºC. Soil carbon is present in the top six layers of the soil column down to a depth of 3.35 m.

(iii)    Permafrost model

A representation of permafrost carbon has also been added to the model. Permafrost carbon is prognostically generated within the model using a diffusion-based scheme meant to approximate the process of cryoturbation (MacDougall & Knutti 2016). In model grid-cells with perennially frozen soil layers soil carbon is diffused proportional to the effective carbon concentration



of each soil layer. Effective carbon concentration is carbon concentration divided by porosity and a saturation factor (MacDougall & Knutti 2016). Carbon that is diffused into perennially frozen soil is reclassified as permafrost carbon and is given different properties from regular soil carbon. Permafrost carbon decays with its own constant decay rate and is subject to an `available fraction', which determines the fraction of permafrost carbon that is available to decay. The available fraction slowly increased if permafrost carbon becomes thawed and decreased if permafrost carbon decays. Using this scheme, the model can represent the large fraction of permafrost carbon that is in the passive soil carbon pool, while still allowing the passive pool to eventually decay (MacDougall & Knutti 2016).

## 2.2. Description of the CMIP6 forcing for the UVic ESCM

Anthropogenic forcing from greenhouse gases (GHGs), stratospheric and tropospheric ozone, aerosols and stratospheric water-vapor from methane oxidation are considered. Natural forcing includes solar and volcanic. All data used in the creation of this dataset can be accessed from Input4Mip on the Earth System Grid Federation (ESGF) unless otherwise specified. In the following we will shortly describe how the input data for our simulations with the University of Victoria Earth system climate model (UVic ESCM) was created.

In the standard CMIP6 configuration, the UVic ESCM is forced with $CO_2$ concentration data (ppm) (Meinshausen et al., 2017) which then calculated the radiative forcing internally. These equations were updated to represent the newest findings from (Etminan et al., 2016). In contrast to that, radiative forcing for non-$CO_2$ greenhouse gases (GHGs) was calculated externally and summed up to be used as an additional model input, using concentration data of 43 GHGs (Meinshausen et al., 2017). We use updated radiative forcing formulations for $CO_2$, $CH_4$ and $N_2O$ following the findings of Etminan et al. (2016). Radiative forcing of other GHGs was calculated using the formulations in Table 8.A.1 from the IPCC AR5 (Shindell et al., 2013). Meinshausen et al. (2017) introduced three options for calculating radiative forcing from GHGs concentrations. For this study we chose to use the option in which one uses specific calculations for all available 43 GHGs, rather than treating some groups of GHGs in a similar manner.

The radiative forcing of stratospheric water-vapor from methane oxidation was calculated following the suggestion from Smith et al. (2018) by multiplying $CH_4$ effective radiative forcing by 12%.

To calculate radiative forcing of tropospheric ozone, $F_{O3tr}$, the equations from Smith et al. (2018) were used:

$$F_{O3tr} = \beta_{CH4}(C_{CH_4} - C_{CH_4,pi}) + \beta_{NO_x}(E_{NO_x} - E_{NO_x,pi}) + \beta_{CO}(E_{CO} - E_{CO,pi}) + \beta_{NMVOC}(E_{NMVOC} - E_{NMVOC,pi}) + f(T) \quad (1)$$

$$f(T) = min\{0, 0.032 * ext(-1.35 * T) - 0.032\} \quad (2)$$

where $\beta$ are transfer coefficients, $C_{CH4}$ are methane concentrations, $E_X$ are emissions of the respective species ($NO_x$ - nitrate aerosols, $CO$ – carbon monoxide, $NMVOC$ - non-methane volatile organic compounds), $E_{X,pi}$ are the respective pre-industrial constants for the specific species, and $T$ is temperature in Kelvin. Note that $f(T)$ was not included in our


calculations because the forcing is not calculated dynamically. Concentrations and emissions data were obtained from

2    Input4Mip on the Earth System Grid Federation. Pre-industrial values were taken from Table 4 from Smith et al. (2018).

Again following Smith et al. (2018), radiative forcing of stratospheric ozone, $F_{O3st}$, can be calculated from GHG

4    concentration data using

$$F_{O3st} = a(bs)^c \tag{3}$$

6    with $s = r_{CFC11} \sum_{i \in ODS}(n_{Cl}(i)Ci\frac{r_i}{r_{CFC11}} + 45 n_{Br}(i)C_i\frac{r_i}{r_{CFC11}})$ (4)

where $a = -1.46E^{-5}$, $b = 2.05E^{-3}$ and $c = 1.03$ are curve fitting parameters, and $r_{CFC11}$ is the fractional release values for

8    Trichlorofluoromethane. Equivalent stratospheric chlorine of all ozone depleting substances (ODS) is represented by Eq. (4)

as a function of ODS concentrations. $r_i$ are fractional release values for each ODS as defined by Daniel and Velders (2011).

10    Halon 1202 data are not provided by Input4Mips and therefore was not included.

Three-dimensional aerosol optical depth (AOD) input for the UVic ESCM was created using a UVic ESCM grid and

12    the scripts and data provided by Stevens et al. (2017), describing nine plumes globally which are scaled with time to produce

monthly sulphate aerosol optical depth forcing for the years 1850-2018 (for comparison see Fig. S1). The resulting AOD

14    caused a too strong forcing in the historical period, a scaling routine was therefore implemented into the UVic ESCM, which

allows to scale the derived albedo alteration and accordingly aerosol forcing from AOD data. For transient simulations, the

16    scaling factor was set to 0.7, which gives a globally average forcing of -1.4296 Wm$^{-2}$ in 2011, consistent with the IPCC AR5

range estimate of between -2.3 and 0.2 Wm$^{-2}$ (Ciais and Sabine, 2013).

18    Anthropogenic land-use changes (LUC) in the UVic ESCM are prescribed from standardized CMIP6 land use forcing

(Ma et al., 2019) that has been re-gridded onto the UVic grid. These gridded land use data products (LUH2), which contain

20    information on multiple types of crop and grazing lands, were adapted for use with UVic by aggregating the crop lands and

grazing lands into a single "crop" type, which can represent any of 5 crop functional types, and a single "grazing" variable,

22    which represents both pasture and rangelands. This forcing is used by the model to determine the fraction of each grid cell

that is crop or grazing land with those fractions of each terrestrial grid cell then assigned to C3 and C4 grasses and excluded

24    from the vegetation competition routine of the TRIFFID component. CO$_2$ emissions from LUC affect the model runs so that

when forest or other vegetation is cleared for crop lands, range lands or pasture, 50% of the carbon stored in trees is released

26    directly in to atmosphere and the remaining 50% is placed into the short-lived carbon pool.

Historical volcanic radiative forcing data is provided by Schmidt et al. (2018). Following CMIP6 spin-up forcing

28    recommendations (Eyring et al., 2016) volcanic forcing is applied as an anomaly relative to the 1850 to 2014 period in the

UVic ESCM.

30    Solar constant data for 1850 to 2300 was accessed from Input4Mips (Matthes et al., 2017). The available monthly

data was annually averaged. Following CMIP6 spin-up forcing recommendations (Eyring et al., 2016) spin-up values were set

32    to the mean of 1850-1873, equal to 1360.7471 Wm$^{-2}$.





A comparison of radiative forcing used for the UVic ESCM for the Coupled Model Intercomparison Projects 5 and 6 (CMIP5 and CMIP6, respectively) and the data for the historical period as given by IMAGE model (Meinshausen et al., 2011) is shown in Fig. S2.

**2.2. Description of the CMIP6 forcing for the UVic ESCM**

The model was spun up with boundary conditions as described in the CMIP6 protocol by Eyring et al. (2016) for over 10,000 years, in which the weathering flux was dynamically simulated and diagnosed. To diagnose the transient climate response (TCR), equilibrium climate sensitivity (ECS), the ocean heat uptake efficiency ($\kappa\_4x$), and the transient climate response to cumulative emissions (TCRE), as given in Table 1) we ran 1,000 year simulations starting with a 1% per year increase in $CO_2$ concentrations until a doubling (2xCO2) and quadrupling (4xCO2) was reached after which the concentration was kept constant. Before switching from $CO_2$ concentration driven simulations to $CO_2$ emissions driven simulations, a 1,500 years drift simulation was run. Finally, the historical simulation is forced with fossil $CO_2$ emissions, dynamically diagnosed land-use change emission, non-$CO_2$ GHG forcing, sulphate aerosol forcing, volcanic anomalies forcing and solar forcing.

**2.3. Fine tuning of the UVic ESCM 2.10**

We tested version 2.10 of the UVic ESCM, with the main incentive to improve its skill in simulating carbon fluxes, historical temperature trajectories and ocean tracers. While evaluating the model with available observational data, specific additional changes and updates were applied with respect to the UVic ESCM versions 2.9-02 (Eby et al., 2009) and 2.9-CE (Keller et al., 2014).

After merging the two model versions, the UVic ESCM's simulated historical cumulative land-use change emissions were close to zero, since its pre-industrial vegetation resembled closely the pattern of plant functional types of today. In order to get a good representation of deforested biomass, we updated the vegetation parameterization to ensure that diagnosed historical land-use change carbon emissions agree with observational estimates from LeQuéré et al. (2018). During this process there was a trade-off between getting the right amount of LUC emissions and a good representation of present day broadleaf trees in tropical areas. In the end the representation of LUC emissions had the higher priority, to be able to simulate emission driven simulations. To slightly mitigate the high broadleaf tree density, we then decreased the terrestrial $CO_2$ fertilization by 30% following Mengis et al. (2018), by adjusting the atmospheric $CO_2$ concentration that is used by the terrestrial model component. This was done to reduce the overestimation of broadleaf tree vegetation especially in tropical areas which, in the real world are limited by phosphorus (Camenzind et al., 2018). The broadleaf tree representation and the terrestrial carbon flux were improved by the scaling of the $CO_2$ fertilization strength (see Sect. 3.1. and 3.2.); the terrestrial carbon fluxes are now in better agreement with the Global Carbon Budget 2018 by (LeQuéré et al., 2018).

The new model version equilibrated with a rather low oceanic overturning strength, we therefore increased the ocean background vertical diffusivity from previous value of 0.15 cm$^2$ s$^{-1}$ in Keller et al. (2014) to 0.25 cm$^2$ s$^{-1}$, to increase ocean





overturning (see Fig. S3 and S4). This caused ocean diffusivity to slightly increase in depths between 0 and 3500 m relative to the previous model version (Fig. S3), but to follow the tidal mixing profile very closely for higher depth. Global diffusivity increased by about 4 %. This change enabled us to reach a very similar ocean overturning as found for the UVic ESCM 2.9-02, which uses the Bryan-Lewis mixing scheme (Fig. S3 and S4). This stronger overturning then in turn also improved ocean physical properties (see Sect. 3.3. and supplementary material) and the global mean temperature and warming trends. However, it also causes the ocean heat content anomaly for the upper 700m to amount to $23.9 \cdot 10^{22}$ J, which is an overestimation of the observed 700m OHC anomaly of $16.7 \pm 1.6 \cdot 10^{22}$ J (Levitus et al., 2012) (Table 1). This seems to be a general feature of EMICs (Eby et al., 2013), but might still be problematic.

## 3. Evaluation of model components

In this section we evaluate the performance of the different components of the UVic ESCM version 2.10 based on observations.

### 3.1. Global key metrics – Temperature, Carbon Cycle, Climate Sensitivity and Radiation Balance

The emission-driven, transient historical climate simulation of the UVic ESCM version 2.10 forced with CMIP6 data reproduces well the historical temperature trend in the 20[th] century of $0.75 \pm 0.21$ K as derived from the Global Warming Index (Haustein et al., 2017) (Table 1, Fig. 1). Starting from the year 2000 the simulated global mean temperature increases at a higher rate than previously, but the total temperature change since preindustrial remains within the uncertainty range of the estimate in the latest IPCC special report on 1.5 °C (Fig. 1, light grey cross) (Rogelj et al., 2018). This steep temperature increase over the last twenty years of simulation amounts to a rate of temperature change of 0.21°C/decade, which is slightly higher than the best estimate from the infilled HadCRUT4-CW dataset of 0.17°C/decade (uncertainty range of 0.13-0.33°C/decade) (Haustein et al., 2017).

The simulated transient climate response (TCR) for a doubling and quadrupling of atmospheric CO2 concentration are with 2.25 °C and 4.83 °C, respectively, on the higher end of the reported ranges from the EMIC comparison study by Eby et al. (2013). This can be partly explained by the updated $CO_2$ forcing formulation that was adopted from Etminan et al. (2016). For an atmospheric $CO_2$ concentration of 1120 ppm (i.e. 4 times pre-industrial $CO_2$) the new formulation gives a forcing of 8.08 W m$^{-2}$, compared to the previous formulation implemented in the UVic ESCM that gave a forcing of 7.42 W m$^{-2}$. In the same way, high values are diagnosed for the Equilibrium Climate Sensitivity for a 2x and 4x increase in atmospheric $CO_2$ concentrations, with temperature increases of 4.08 °C and 7.07 °C, respectively. The value for ECS_2x even lies outside the EMIC range as found by Eby et al. (2013). This might be caused by the rapid drop in Southern Hemisphere sea ice cover after about 430 years / 500 years in the 2x and 4x CO2 simulations, which in turn decreases the surface albedo and therefore causes more warming. Since the ocean heat uptake efficiency is assessed at year 140 of the TCR4x simulation, it is as the TCR4x with 0.98 W m$^{-2}$ °C$^{-1}$ on the higher end, but still within the EMIC range, and does not yet reflect this process. In the same way,





the transient climate response to cumulative emissions (TCRE) is with 2.2 K (1000 PgC)$^{-1}$ higher than in previous model
version but still within the likely range reported by the IPCC AR5 (Table 1).

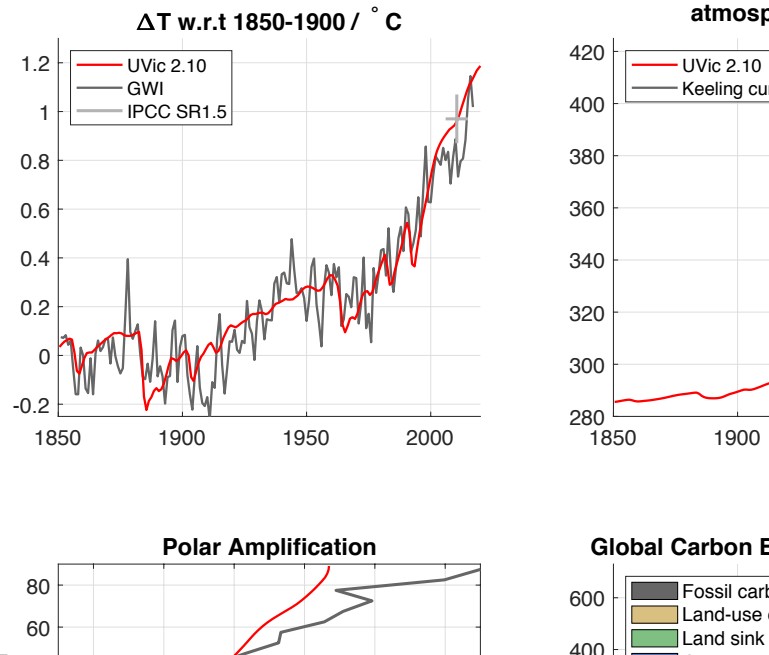

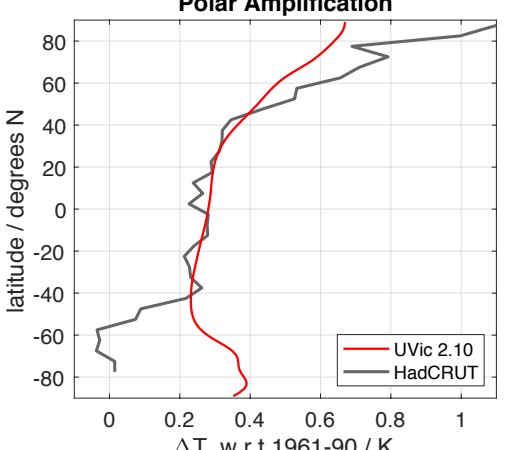

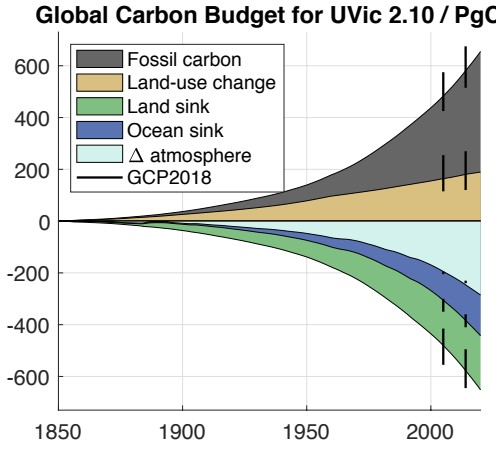

**Figure 1: (a) Global mean temperature change for the UVic ESCM 2.10 relative to 1850-1900 (red line) in comparison with the average observed warming using the Global Warming Index dataset from (Haustein et al., 2017) (grey line) and the IPCC's special report on 1.5C GSAT temperature change for 2006-2015 (light grey cross) (b) Atmospheric CO2 concentrations in the UVic ESCM 2.10 (red line) in comparison with the Keeling curve from the Mauna Loa observatory (Keeling et al., 2005; grey line) (c) Zonal means of temperature change of the HadCRUT median near surface temperature anomaly (grey line) (Morice et al., 2012) in comparison to the UVic ESCM 2.10. All temperature changes are for a 30-years mean around 1995 with respect to the 1961-1990 period in K. (d) The global carbon budget for the UVic ESCM 2.10 partitioned into fossil fuel carbon and land-use carbon emissions and atmosphere, land and ocean sinks, compared to cumulative carbon fluxes between 1850 and 2005 and 1850 and 2015 from the Global Carbon Project 2018 (grey lines) (LeQuéré et al., 2018).**





**Table 1: Key global mean metrics of the UVic ESCM 2.10 compared to relevant observations or model intercomparison projects. dT_20th century is the change in surface air temperature over the 20th century from the historical "all" forcing experiment. TCR_2xCO2, TCR_4xCO2, and ECS_4xCO2 are the changes in global average model surface air temperature from the decades centered at years 70, 140, and 995 respectively, from the idealized 1 % increase to 4xCO2 experiment. The ocean heat uptake efficiency, $\kappa$_4x, is calculated from the global average heat flux divided by TCR_4xCO2 for the decade centered at year 140, from the same idealized experiment. Note that ECS_4xCO2 was calculated from the decade centered at year 995 from the idealized 1 % increase to 2xCO2 experiment.**

| | UVic ESCM 2.10 | comparison data | |
| --- | --- | --- | --- |
| | | values | citation |
| dT_20th century - global | 0.74 °C | 0.75 ± 0.21 °C <br> 0.78 [0.38 – 1.15] °C | Haustein et al., 2017 <br> EMIC range - Eby et al., 2013 |
|          - ocean <br>          - land | 0.72 °C <br> 0.78 °C | | |
| TCR_2xCO2 | 2.25 °C | 1.8 [0.8 – 2.5] °C <br> 1.8 ± 0.6 °C | EMIC range - Eby et al., 2013 <br> CMIP5 range - IPCC AR5 WG1 |
| TCR_4xCO2 | 4.83 °C | 4.0 [2.1 – 5.4] °C | EMIC range - Eby et al., 2013 |
| ECS_2xCO2 | 4.08 °C | 3.0 [1.9 – 4.0] °C <br> 3.2 ± 1.3 °C | EMIC range - Eby et al., 2013 <br> CMIP5 range - IPCC AR5 WG1 |
| ECS_4xCO2 | 7.07 °C | 5.6 [3.5 – 8.0] °C | EMIC range - Eby et al., 2013 |
| $\kappa$_4x | 0.98 W m$^{-2}$ °C$^{-1}$ | 0.8 [0.5 – 1.2] W m$^{-2}$ °C$^{-1}$ | EMIC range - Eby et al., 2013 |
| TCRE | 2.20 K (1000PgC)$^{-1}$ | 0.8 – 2.5 K (1000PgC)$^{-1}$ | IPCC AR5 SPM |
| NH sea ice area (sept) | 2005-15: 3.4 million km$^2$ <br> -0.24 million km$^2$ decade$^{-1}$ | 5.5 [3 -10] million km$^2$ <br> -1.07 to -0.73 million km$^2$ dec$^{-1}$ <br> (1979-2012) | CMIP5 - Stroeve et al., 2012 <br> IPCC AR5, Chp. 4 |
| SH sea ice area (feb) | 2005-15: 1.3 million km$^2$ <br> -0.25 million km$^2$ dec$^{-1}$ | 1979-2010: 3.1 million km$^2$ <br> 0.13 to 0.2 million km$^2$ dec$^{-1}$ | IPCC AR5, Chp. 4, <br> Parkinson and Cavalieri, 2012 |
| Precipitation    - global <br>          - ocean <br>          - land | 1060 mm <br> 1167 mm <br> 814 mm | 818 mm | Hulme et al., 1998 |
| dPrecip.       - global <br>          - ocean <br>          - land | -0.41 mm dec$^{-1}$ <br> -0.1 mm dec$^{-1}$ <br> -0.43 mm dec$^{-1}$ <br> -1.2 mm dec$^{-1}$ | [-4.2 – 1.2] mm dec$^{-1}$ <br> [-7 – 2] mm dec$^{-1}$ | CMIP5 - Kumar et al., 2013 <br> obs - IPCC AR4 |
| NH permafrost area | 17.0 million km$^2$ | 18.7 million km$^2$ | Brown et al., 1997; <br> Tarnocai et al., 2009 |
| overturning    - AMOC <br> <br>          - AABW | 17.5 Sv <br> <br> -8.5 Sv | 17.6 ± 3.1 Sv <br> 18.7 ± 4.8 Sv <br> -5.6 ± 3.0 Sv | Lumpkin and Speer, 2007 <br> Rayner et al., 2011 |
| ocean surface pH | 2005: 8.07 | ~8.1 | IPCC AR5, Fig. 6.28 |
| ocean heat content anomaly      0-700m <br>                 0-2000m | 23.9 10$^{22}$ J <br> 36.3 10$^{22}$ J | 16.7 ± 1.6 10$^{22}$ J <br> 24.0 ± 1.9 10$^{22}$ J | Levitus et al., 2012 |

(TCR – Transient Climate Response, ECS – Equilibrium climate response, $\kappa$_4x – ocean heat uptake efficiency, TCRE – Transient Climate Response to Cumulative Carbon Emissions, NH – Northern hemisphere, SH – southern hemisphere, AMOC – Atlantic Meridional Overturning, AABW – Antarctic Bottom Water)





**Table 2: Global carbon cycle fluxes for year 2005 in PgC yr$^{-1}$ (- flux) or cumulated fluxes between 1850 and 2005 in PgC (- cum) from the UVic ESCM 2.10 compared to data-based estimates from the Global Carbon Project 2018 and the IPCC AR5 Chp.6. Note the observational estimates of the carbon stocks are calculated from 1750 to 2005.**

| | | UVic 2.10 | comparison data | |
| --- | --- | --- | --- | --- |
| | | | values | citation |
| **Fossil fuel** | - cum | 323 | 320 ± 15 | LeQuéré et al., 2018 |
| | - flux | 6.5 | 7.8 ± 0.4 | |
| **Land-use change** | - cum | 165 | 185 ± 70 | LeQuéré et al., 2018 |
| | - flux | 1.6 | 1.3 ± 0.7 | |
| **Change in atmos. C** | - cum | 197 | 200 ± 5 | LeQuéré et al., 2018 |
| | - flux | 4.7 | 4.0 ± 0.02 | |
| **Land carbon sink** | - cum | 175 | 160 ±45 | LeQuéré et al., 2018 |
| | - flux | 1.8 | 2.7 ± 0.7 | |
| **Land gross primary production** | | 146 PgC yr$^{-1}$ | 123 ± 8 PgC yr$^{-1}$ | Beer et al., 2010 Ciais et al., 2013 |
| **Ocean carbon sink** | - cum | 113 | 125 ± 20 | LeQuéré et al., 2018 |
| | - flux | 2.4 | 2.1 ± 0.5 | |
| **Ocean net primary production** | | 70 PgC yr$^{-1}$ | 44-67 PgC yr$^{-1}$ | Behrenfeld et al., 2005; Westberry et al., 2008 |
| **NH permafrost carbon** | | 497 | ~500 | Hugelius et al., 2014 |
| **Permafrost affected soil carbon** | | 1009 | 1035 ± 150 | Hugelius et al., 2014 |

Overall, the global carbon-cycle fluxes of the UVic ESCM 2.10 are within the uncertainty ranges of the Global Carbon Project (LeQuéré et al., 2018, GCP18) (Table 2, Fig. 1). The $CO_2$ concentrations as simulated in the emissions driven simulation follow the Keeling curve closely, but there is a slightly higher increase between 1960 and 2010 in the simulation with an increase of 75 ppm than in the observations with 73 ppm. The change in atmospheric carbon between 1850 and 2005 is however within the uncertainty estimate of the GCP18 (Table 2). The land-use change emissions which are generated dynamically in the model by changes in agriculturally used areas, reach a cumulative level of 165 PgC between 1850 and 2005 and are hence well within the uncertainty range of the GCP18 estimate of 185 ± 70 PgC (Table 2). Both the cumulative ocean sink with 113 PgC and the land sink with 175 PgC in the period between 1850 and 2005 are within the uncertainty range of the GCP18 (Table 2). While the land sink is slightly higher than the best estimates the ocean sink is at the lower end of the given range.





**Table 3: Global radiation balance of the UVic ESCM 2.10 in comparison with Wild et al. (2013), shown are unitless albedo values and radiation fluxes in units of W m⁻².**

| | UVic 2.10 | Wild et al., 2013 | |
| --- | --- | --- | --- |
| | 2000-2010 | Observations 2001-2010 | CMIP5 range 1985-2004 |
| TOA Solar down | 340 | 340 (240,341) | (338.9,341.6) |
| TOA Solar up | 104 | 100 (96,100) | (96.3, 107.8) |
| Planetary albedo | 0.305 | 0.294 | 0.300 |
| TOA Solar net | 237 | 240 | (233.8, 244.7) |
| TOA Thermal up | 235 | 239 (236,242) | (232.4, 243.4) |
| solar absorbed atmos. | 68 | 79 (74,91) | (69.7, 79.1) |
| Surface solar down | 203 | 185 (179,189) | (181.9, 197.4) |
| Atmospheric albedo | 0.227 | 0.250 | 0.255 |
| Surface solar up | 35 | 24 (22,26) | (20.9, 31.5) |
| Surface albedo | 0.171 | 0.130 | 0.131 |
| Surface solar net | 168 | 161 (154,166) | (159.6, 170.1) |
| Surface thermal net | -51 | -55 | (-65.2, -49.4) |
| Surface Latent heat | 76 | 85 (80,90) | (78.8, 92.9) |
| Surface Sensible heat | 31 | 20 (15,25) | (14.5, 27.7) |

(TOA – top of the atmosphere)

The simulated top of the atmosphere (TOA) short wave and long wave radiation of the UVic ESCM for the year 2005 lies well within the range of the CMIP5 models as reported by Wild et al. (2013) and agrees reasonably well with the observed estimates for both the solar and the thermal radiation fluxes (Table 3 or Fig. S5). The same is true for the simulated net surface thermal flux, which is with -51 W m⁻², at the lower end of the CMIP5 range (Table 3). Now following the solar radiation through the energy moisture balance model, however, we find that the simulated atmospheric albedo with 0.227 is comparatively low to the observed value of 0.250. This causes a rather low simulated absorption of solar radiation by the atmosphere of 68 W m⁻², where the observed estimate from Wild et al. (2013) is 79 W m⁻². Thanks to a rather high simulated surface albedo (0.171 compared to 0.13 from observations) the resulting absorbed solar radiation at the surface is still high but in contrast to the atmospheric absorption within the CMIP5 range. This results in a global mean surface net radiation of 117 W m⁻² which is rather high compared to the observed best estimate of 106 W m⁻². This is the radiative energy available at the surface to be redistributed amongst the non-radiative surface energy balance components. Accordingly, the simulated sensible heat flux in the UVic ESCM is with 31 W m⁻² also too high compared to the CMIP5 range of (14.5, 27.7) W m⁻². Finally, the



latent heat flux calculated from simulated evaporation is with 76 W m$^{-2}$ on the very low end of observational and CMIP5 estimates, which is likely linked to the high transpiration sensitivity of plants in the UVic ESCM.

### 3.2. Spatially resolved atmospheric and land-surface metrics

The simulated polar amplification of the UVic ESCM 2.10 compares well to the HadCRUT near surface temperature anomaly data for all latitudes, but the Southern Ocean south of 40°S (Fig. 1). Here the UVic ESCM 2.10 shows more of a warming trend than what is observed. Previous studies have shown that this warming is connected to the representation (or the lack thereof) of poleward intensified winds (Fyfe et al., 2007). This warming trend was already evident in previous versions of the UVic as well as in other Earth system model of intermediate complexity (EMICs) (see Fig. S6 and Fig. 4 in Eby et al., 2013). While the pattern of the seasonal cycle concerning surface air temperature agrees well with the CRU Global 1961-1990 Mean Monthly Surface Temperature Climatology (Fig. 2), the magnitude especially in the northern hemisphere land areas is substantially lower by up to 25 °C.

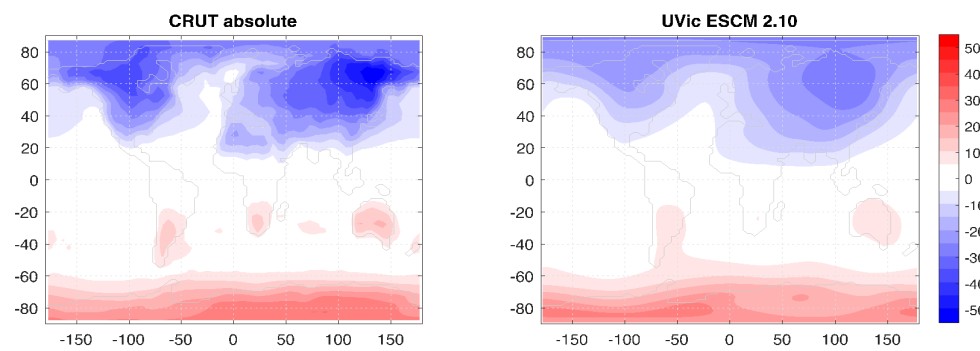

**Figure 2: Seasonality of surface air temperature, as differences between December-January-February and June-July-August means for the CRU Global 1961-1990 Mean Monthly Surface Temperature Climatology (Jones et al., 1999), and the UVic ESCM for the period of 2000-2005.**

The simulated northern hemisphere summer sea ice extent with 3.4 million km$^{-2}$ at the lower end of the CMIP5 estimates, and considerably smaller than the observed sea ice concentration (Table 1, and Fig. 3). This lower extent seems to be mainly due to a lack of simulates summer sea ice concentrations between 15 and 60 %, whereas higher concentrations show a good agreement with the observed pattern (Fig. 3). The southward extension of winter sea ice concentration in the UVic is considerably smaller than the observations from passive microwave satellite missions. Concerning NH summer sea ice trends, the UVic ESCM shows lower trends of -0.24 million km$^{-2}$ dec$^{-1}$ during the last 30 years, compared to what is observed (-1.07 to -0.73 million km$^{-2}$ dec$^{-1}$) (IPCC AR5, Chp.4, Ciais and Sabine, 2013). The summer sea ice extent in the Southern Hemisphere is with 1.3 million km$^{-2}$ also smaller than the observed 3.1 million km$^{-2}$, and in contrast to the observed increasing trends in sea ice shows a decline of -0.25 million km$^2$ dec$^{-1}$ (Table 1). While this is consistent with the simulated warming trend in the Southern Hemisphere surface air temperature, this is still a bias in the model.



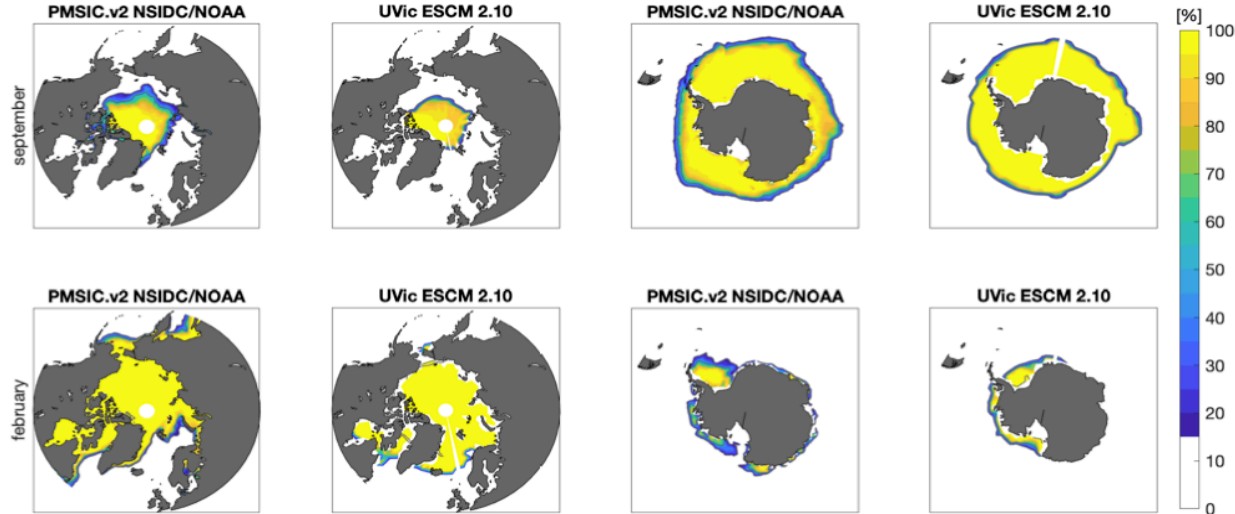

**Figure 3: September (top row) and February (bottom row) sea ice concentration from passive microwave observations (Meier et al., 2013) and the UVic ESCM 2.10 for the northern and southern hemisphere for the period of 2003-2013 in %.**

Observed global mean terrestrial precipitation between 1961-1990 amounts to 818mm (Hulme et al., 1998). The adjusted CO2 fertilization strength in the UVic ESCM 2.10 results in global mean terrestrial precipitation of 814mm for the same period (Table 1), bringing it close to the observed amount. Concerning terrestrial precipitation trends the UVic ESCM 2.10 shows a negative trend in terrestrial precipitation of -0.43 mm decade$^{-1}$ for the period between 1930 to 2004 (Table 1). This is in agreement with the range of terrestrial precipitation trends of [-4.2 – 1.2] mm decade$^{-1}$ given by Kumar et al. (2013). The simulated pattern of annual mean precipitation flux for the last 30 years generally agrees well with the observed pattern (Fig. 4). Similar to the seasonal temperature maps, the UVic ESCM slightly underestimates the most extreme amplitudes of annual mean precipitation located in the tropical areas. The latitudinal mean values agree well in magnitude, but the tropical rain bands are extending too far north and south. Total terrestrial precipitation with 814 mm agrees well with the observed 1961-1990 mean value of 818 mm (Hulme et al., 1998). The terrestrial precipitation trend of -1.2 mm per decade also agrees well with the observed terrestrial precipitation changes for the recent historical period (1951 to 2005) of −7 to +2 mm per decade, with error bars ranging 3–5 mm per decade (IPCC AR4) (Table 1).



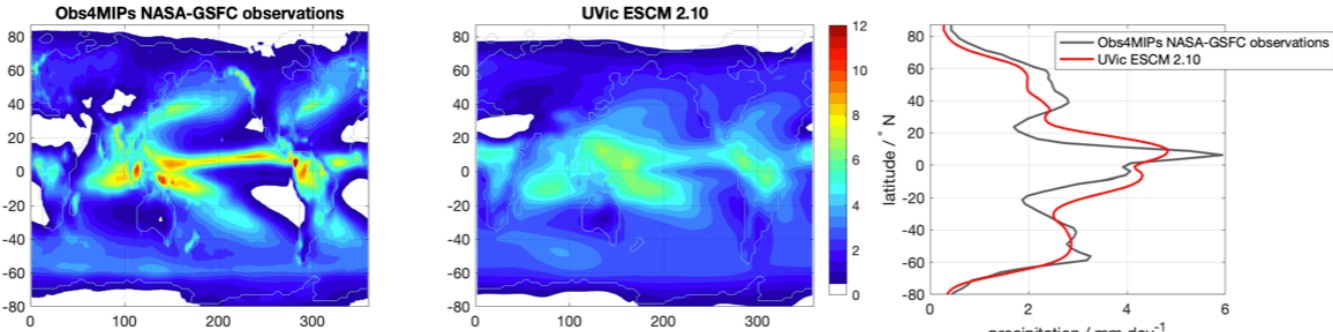

**Figure 4: Mean precipitation flux for the period 1979-2013 in units of mm day⁻¹ from Obs4MIP (Adler et al., 2003) (left, and grey line) and the UVic ESCM 2.10 (middle, and red line), and zonally averaged values as a function of latitude (right).**

The simulated air-sea carbon flux for 2000 to 2010 agrees with observations from Takahashi et al. (2009) (Fig. 5). Oceanic carbon uptake takes place at high latitudes and carbon is mainly released in the tropical Pacific. In the Southern Ocean observations show slightly positive values (i.e. carbon being released to the atmosphere) which are not reproduced by the UVic ESCM 2.10. This is also evident in the latitudinal means, where the UVic ESCM 2.10 generally shows good agreement with the observations, but simulates ocean carbon uptake south of 50 °S, where the observations show low or slightly positive air to sea carbon fluxes.

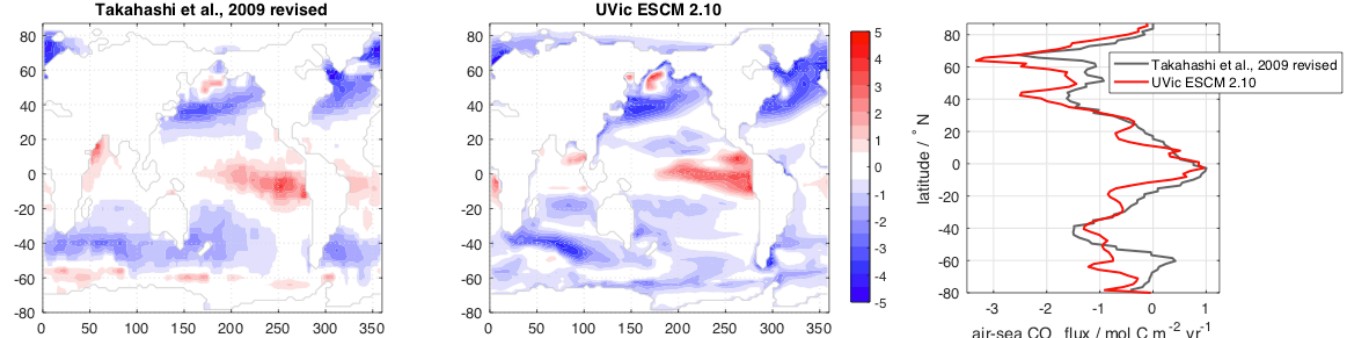

**Figure 5: Air - sea carbon flux for the year 2005 in units of mol C m⁻² yr⁻¹ from the revised dataset from Takahashi et al. (2009) (left, and grey line) and the UVic ESCM 2.10 (middle, and red line), and zonally averaged values as a function of latitude (right).**

The UVic ESCM overestimates vegetation carbon density of in tropical rainforest regions, such as in South America, and Central Africa when compared to the revised estimates of Olson (1983,1985,2001) (Fig. 6). More recent biomass studies have challenged Olson's estimates for some regions of the world, but Olson (1983,1985) still provides the only globally-consistent estimate of global carbon stored in vegetation. This positive bias in the UVic ESCM 2.10 in the tropics is due to an overestimation of broadleaf trees, which is the plant functional type with the highest carbon density in the UVic ESCM (see Fig. S7). This overestimation of broadleaf trees leads to a small overestimation of global mean gross primary production in 2005 on land, 146 PgC yr⁻¹, compared to the observation-based estimate of 123 ± 8 PgC yr⁻¹ using eddy covariance flux data





and various diagnostic models (Beer et al., 2010) (Table 2). In contrast, the simulated vegetation coverage of carbon densities of 2-5 kgC m$^{-2}$ is lower than observations especially in Central Asia and at higher northern latitudes. This, however, does not imply that the dominant plant functional types, namely C3/C4 grasses, are underrepresented in this area. In the UVic ESCM 2.10 the representation of C3/C4 grasses, as well as needleleaf trees, in high northern latitudes improved compared to earlier versions (see Fig. S7) thanks to the more complex soil module and the corresponding vegetation tuning. In summary, the UVic ESCM overestimates broadleaf tree cover in the tropics, but improved the representation of the vegetation cover at latitudes north of 20 °N compared to previous model versions.

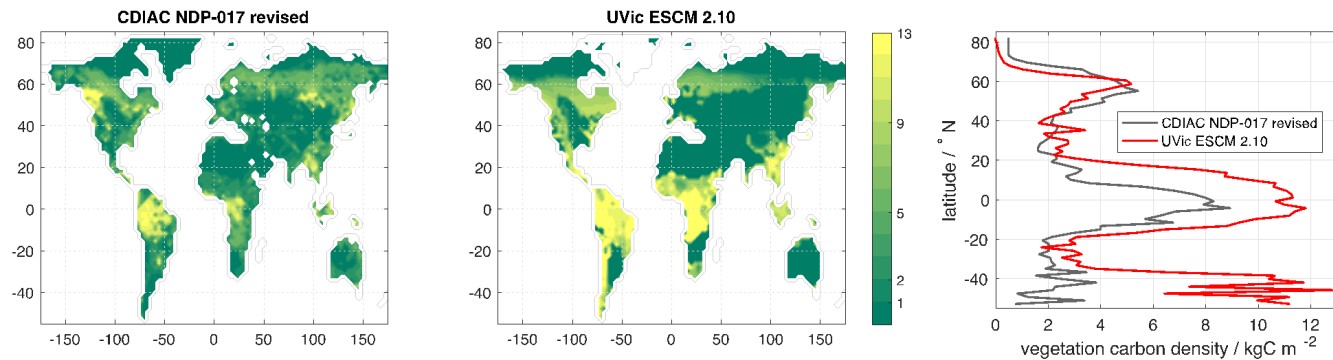

**Figure 6: Vegetation carbon density for the 1960-2000 period in units of kg C m$^{-2}$ from the revised CDIAC NDP-017 dataset (Olson et al., 2001) (left, and grey line) and the UVic ESCM 2.10 (middle, and red line), and zonally averaged values as a function of latitude (right).**

Simulated soil carbon densities at high northern latitudes compare reasonably well with the map of permafrost soil carbon based on observations by Hugelius et al. (2014) (Fig. 7). While there are regional biases especially in Eastern Canada, the simulated carbon densities in the permafrost areas do have the correct order of magnitude. The total global permafrost carbon of 497 PgC and the total soil carbon in the permafrost region of 1009 PgC, agree well with the reported ~500 PgC and 1035 ± 150 PgC, respectively (Hugelius et al., 2014). The simulated permafrost area is limited to about 60 °N and does not extend as far south as what is observed.



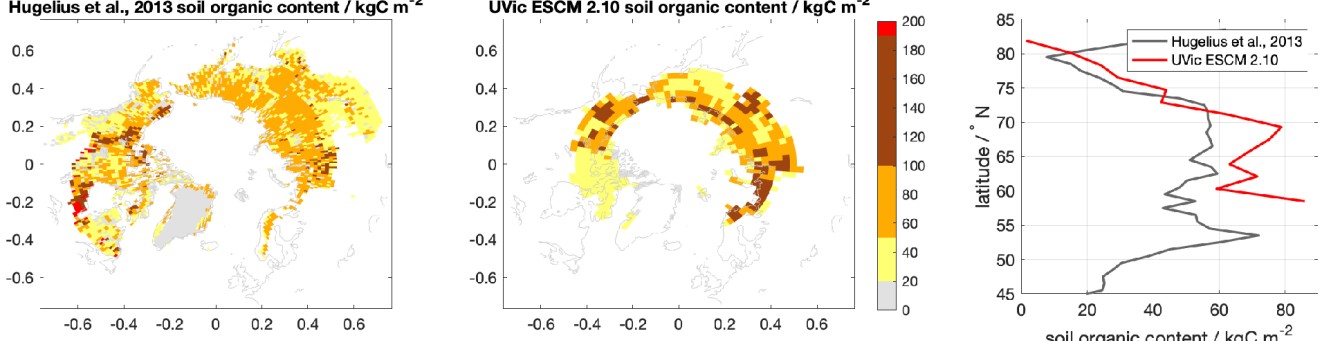

**Figure 7: Soil organic carbon content in permafrost affected soils for the 1980-2000 period in the top 3m of soil in units of kg C m⁻² from the dataset by Hugelius et al. (2014) and for the UVic ESCM 2.10, and zonally averaged values as a function of latitude (right).**

Smith and Burgess (2002) provide a dataset of permafrost depth observations for Canada based on temperature readings, which is a compilation of borehole data across Canada ranging in observation date from between 1966 to 1990. Each borehole is a single observed value, this compares the simulation to a snapshot in time, rather than a temporal average. Permafrost depth in the observational dataset was determined based on the bottom boundary identified by the temperature gradient to be below 0 °C. Permafrost depth distribution in North America simulated by the UVic ESCM broadly agrees with the observed distribution (Fig. 8). The UVic ESCM 2.10 simulates permafrost thicknesses of up to 250 m all around the Arctic circle. Recall that the depth of the UVic ESCM is limited to 250 m and that the vertical resolution is coarser at deeper soil layers. As already seen for the soil organic carbon content the simulated permafrost areas do not extent as far south as what is observed. However, for the purpose of this comparison, the scale for observed permafrost depths was limited to 250 m, but actually many observations show deeper PF thicknesses.

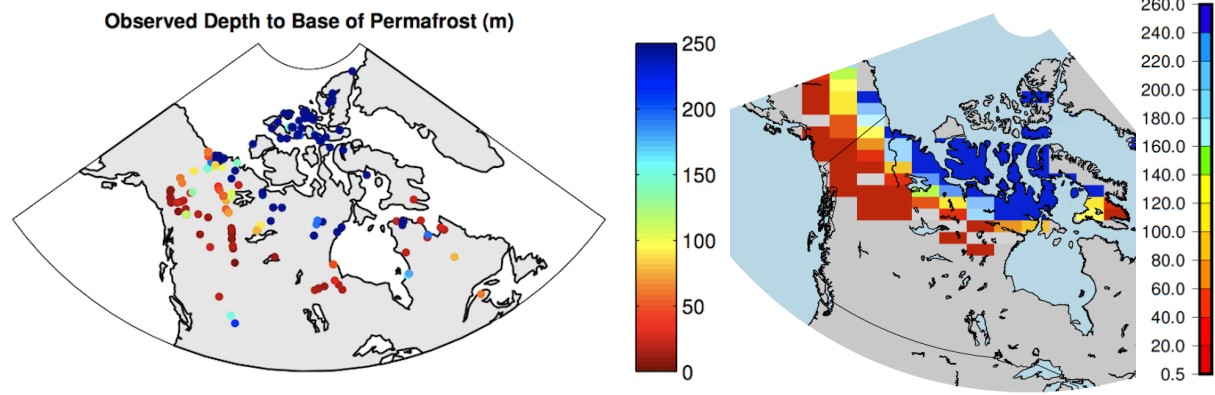

**Figure 8: Observed depth of permafrost for the region of Northern Canada (left) (data source: Smith and Burgess, 2002, figure source: Avis (2012)), the colour bar has been restricted to 250 m depth to aid in comparison despite the fact many locations are deeper, (Avis, 2012); simulated mean permafrost depth for 1966-1990 of the UVic ESCM 2.10 (right).**



### 3.3. Ocean metrics – physical and biogeochemical

In the following section we will compare simulated ocean metrics with observations from the World Ocean Atlas 2018 (WOA18) (Locarnini et al., 2019; Zweng et al., 2019; Garcia et al., 2018a; Garcia et al., 2018b) and the Global Ocean Data Analysis Project (GLODAP) and the new mapped climatologies version 2 (Key et al., 2004; Lauvset et al., 2016) for the period of 1980 to 2010.

The Taylor diagram for eight different ocean metrics illustrates that the UVic ESCM 2.10 improves ocean $\Delta$C14, and slightly improves in ocean temperature, salinity, nitrate and phosphate distributions (dots in Fig. 9), relative to the UVic ESCM 2.9 (crosses in Fig. 9), given the same forcing. In contrast, mainly ocean alkalinity, but also dissolved inorganic carbon and oxygen, show either a larger deviation or lower correlation compared to observations than the previous model version. Generally, the model demonstrates skill in simulating these ocean properties, with correlation coefficients higher than 0.9 for all but the salinity and alkalinity fields, and root mean square deviation (rmsd) of below 50% of the global standard deviation of the observations, again with the exception of salinity and alkalinity. In the following we will discuss these features in more detail.

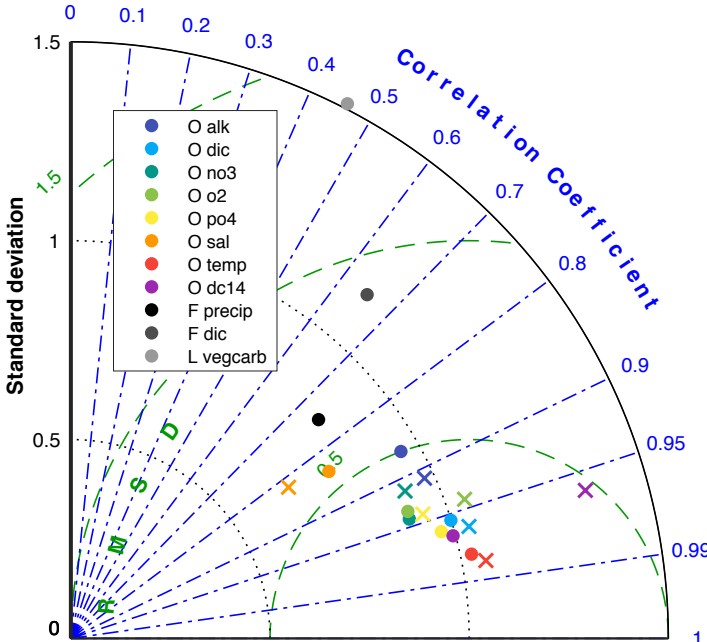

**Figure 9: Taylor diagram (Taylor, 2001) of multiple global UVic ESCM 2.10 fields (dots) and the UVic ESCM 2.9 fields (x) with respect to re-gridded observations from the World Ocean Atlas 2018 (Locarnini et al., 2019; Zweng et al., 2019; Garcia et al., 2018a; Garcia et al., 2018b), GLODAP and GLODAP Mapped climatologies v2 2016b (Key et al., 2004; Lauvset et al., 2016), NASA-GSFC precipitation (Adler et al., 2003), air-sea gas fluxes from Takahashi et al. (2009) and vegetation carbon data from CDIAC NDP-017 dataset (Olson et al., 2001). All datasets are normalized by the standard deviation of the observations. A perfect model with zero rmsd, correlation coefficient of 1, and normalized standard deviation of 1 would plot at (1,0).**

**Geoscientific Model Development** Discussions — EGU Open Access

The vertical profiles of simulated temperature (temp), phosphate (PO4), nitrate (NO3), **Δ**C14, dissolved inorganic carbon (dic) and alkalinity (alk) agree well in magnitude and shape with the observed profiles for all ocean basins and the global ocean (Fig. 10). The general good agreement for these ocean metrics is also seen in comparisons with vertical ocean section and ocean surface maps of these simulated fields with observations (see supplementary material). The only noteworthy biases are too low **Δ**C14 in the central Indian Ocean basin, indicating a too low overturning rate in this ocean basin (see Fig. 10 and 11), and small biases in simulated nitrate showing too high values in the Arctic Ocean compared to observation and too low values in the Indian Ocean (see Fig. 10, Fig. 12 and supplementary material).

**Figure 10: Global and basin-wide averaged vertical profiles of multiple UVic ESCM 2.10 metrics (red lines) compared to observations from the World Ocean Atlas 2018 (Locarnini et al., 2019; Zweng et al., 2019; Garcia et al., 2018a; Garcia et al., 2018b) and GLODAP and GLODAP Mapped climatologies v2 2016b (Key et al., 2004; Lauvset et al., 2016) including standard errors (black solid and dashed lines, respectively) and their respective global misfit (last row) for the period of 1980-2010. Note that for salinity we excluded the fresh water masses observed in the Arctic Ocean (removed all values north of 70 °N for both data sets).**



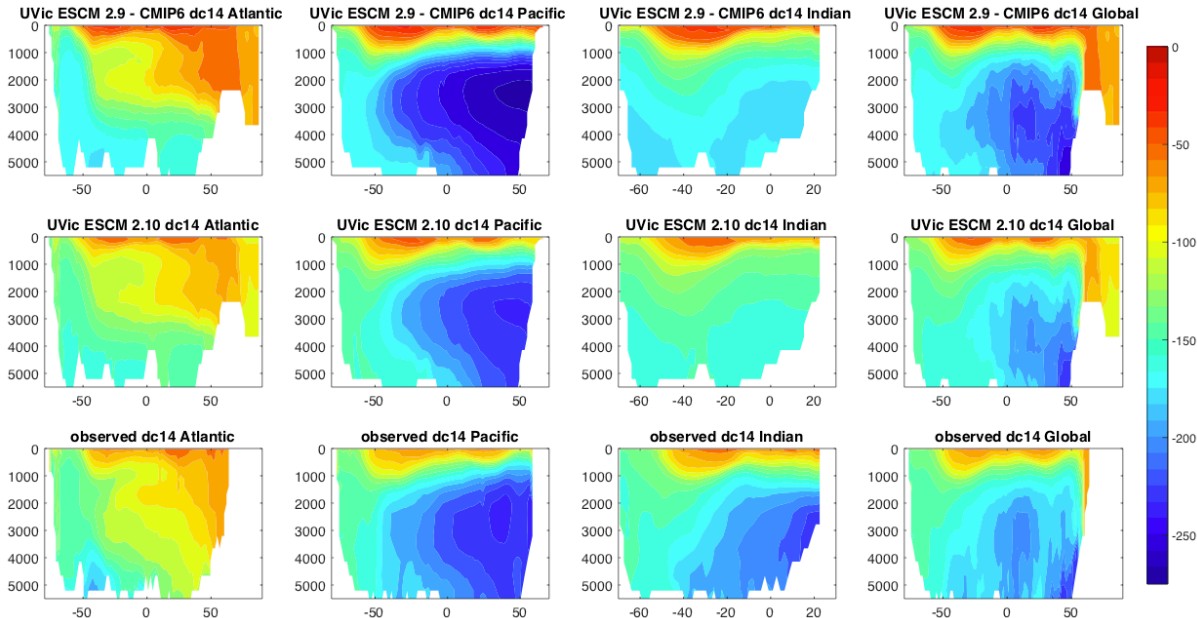

Figure 11: Ocean section of ΔC14 in units of permil for the Atlantic Ocean including the Arctic Ocean (left column), the Pacific Ocean (middle left column), the Indian Ocean (middle right column) and the global average (left column) compared to observations (Key et al., 2004). From top to bottom are shown the published UVic ESCM version 2.9 by Eby et al. (2013) spun-up and forced with CMIP6 forcing, the UVic ESCM version 2.10, both as a mean of the periods 1980-2010 and the observed ocean sections.

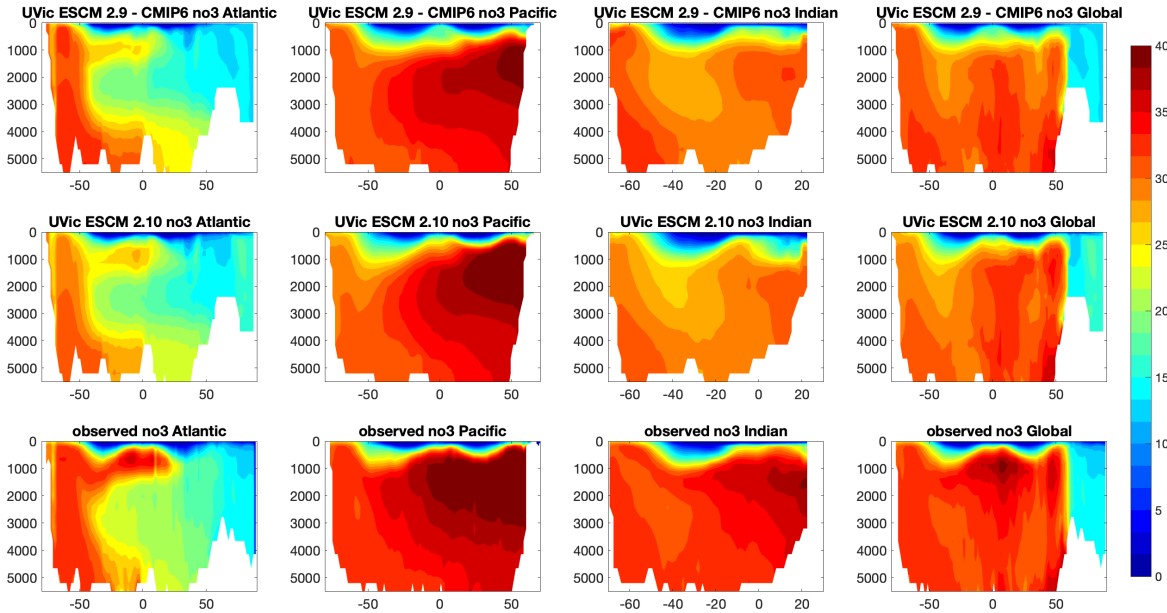

Figure 12: Ocean section of NO3 in units of μmol kg$^{-1}$ for the Atlantic Ocean including the Arctic Ocean (left column), the Pacific Ocean (middle left column), the Indian Ocean (middle right column) and the global average (left column) compared to World Ocean Atlas 2018 (Garcia et al., 2019). From top to bottom are shown the published UVic ESCM version 2.9 by Eby et al. (2013) spun-up and forced with CMIP6 forcing, the UVic ESCM version 2.10, both as a mean of the periods 1980-2010 and the observed ocean sections.

2    To compare ocean salinity profiles (second column Fig. 10) we removed values in the high northern latitudes, north of 70 °N for all regarded datasets. This substantially improved the comparison for the Atlantic Ocean (which in the partitioning

4    includes the Arctic Ocean, see supplementary material), since the UVic ESCM 2.10 does not reproduce well the recent freshening trend associated with sea ice loss and seasonal melt in the Arctic Ocean. The maps of sea surface salinity clearly

6    show this freshening trend (Fig. 13), which also extends to the Pacific Ocean and is hence evident in the vertical profile (Fig. 10). Furthermore, the UVic ESCM 2.10 does not reproduce well the salinity properties of Antarctic Intermediate Water

8    (AAIW), which can be seen by the lack or insufficient representation of the local minimum in about 900 m depth for the global and Atlantic profile (see supplementary material for latitudinal average sections). Note that, the global mean temperature misfit

10   shows similar patterns (Fig. 10). The bias in AAIW salinity in the UVic ESCM 2.10 is caused by too salty surface waters extending southward into the Southern Ocean regions, in which the water is subducted.

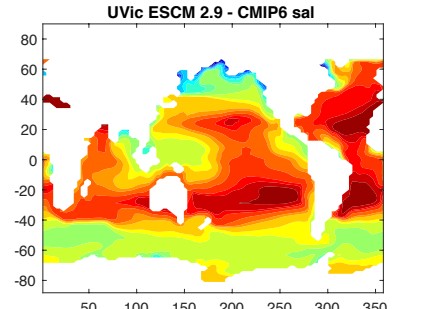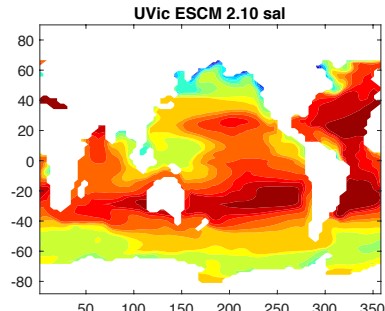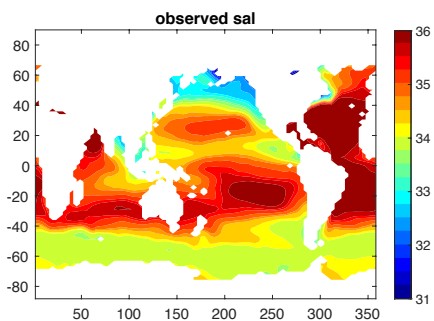

14   **Figure 13: Maps of sea surface salinity in units of psi for the published UVic ESCM version 2.9 (Eby et al., 2013) spun-up and forced with CMIP6 forcing, for the UVic ESCM 2.10, and for the World Ocean Atlas 2018 (Zweng et al., 2019).**

The apparent oxygen utilization (AOU) shows lower values (~15% lower) in the deep ocean, but otherwise reproduces the

18   shape including the local maximum around 1000 m depth well (Fig. 10 and 14). The main bias in AOU is found in the Southern Ocean, where the UVic ESCM 2.9 simulated too high values the UVic ESCM 2.10 now simulated too low oxygen utilization.

20   This is especially true for the Atlantic and Indian Oceans. These biases in AOU are probably linked to biases in the ocean biogeochemistry in the Southern Ocean. Comparing the simulated oxygen minimum zones (OMZ) of the UVic ESCM 2.9 and

22   UVic ESCM 2.10 to observed OMZs, there is an improved representation of the asymmetry of the Pacific OMZ in the newer model version, as well as a reduced bias in the Indian Ocean (Fig. 15).



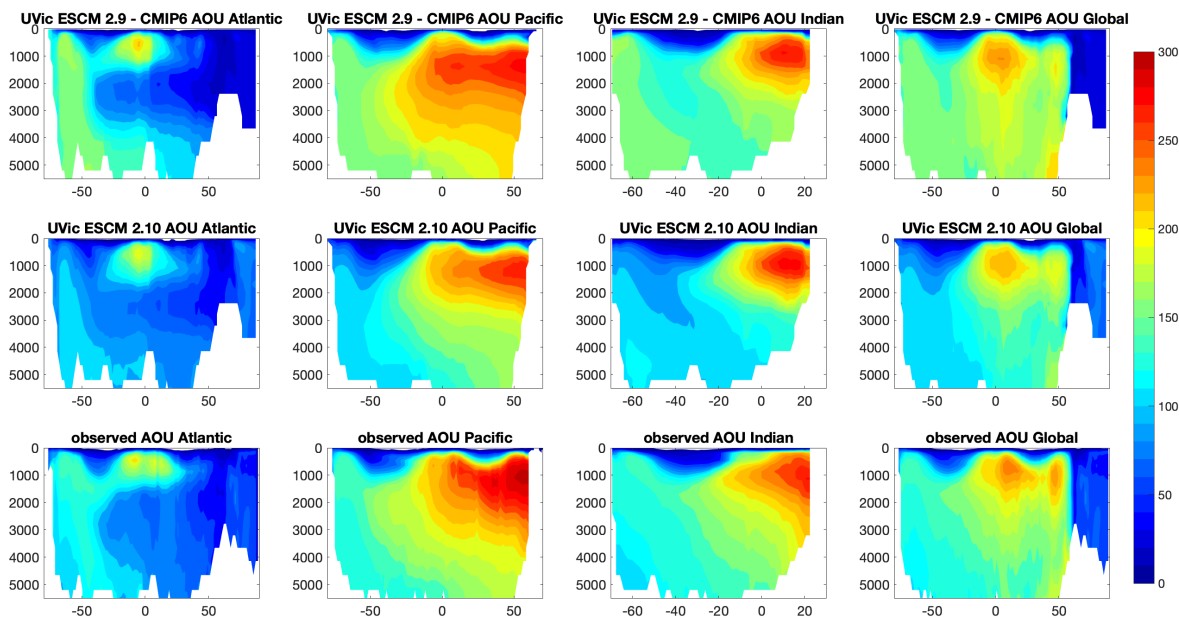

Figure 14: Ocean section of apparent oxygen utilization (AOU) in units of µmol kg⁻¹ for the Atlantic Ocean including the Arctic Ocean (left column), the Pacific Ocean (middle left column), the Indian Ocean (middle right column) and the global average (left column) compared to World Ocean Atlas 2018 (Garcia et al., 2019). From top to bottom are shown the published UVic ESCM version 2.9 by Eby et al. (2013) spun-up and forced with CMIP6 forcing, the UVic ESCM version 2.10, both as a mean of the periods 1980-2010 and the observed ocean sections.

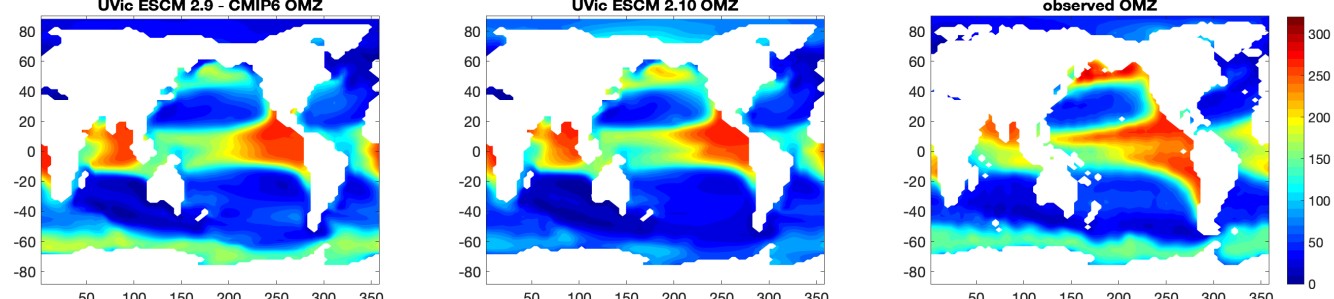

Figure 15: Maps of apparent oxygen utilization in approx. 300 m depth (i.e. the depth of oxygen minimum zones, OMZ) in units of µmol kg⁻¹ for the published UVic ESCM version 2.9-02 (Eby et al., 2013) spun-up and forced with CMIP6 forcing, for the UVic ESCM 2.10, and for the World Ocean Atlas 2018 (Garcia et al., 2019).

## 4. Summary, Conclusion and Outlook

In order to obtain a new version of the University of Victoria Earth System Climate Model (UVic ESCM) that is to be used in the 6th phase of the coupled model intercomparison project (CMIP6), we have merged previous versions of the UVic ESCM to bring together the ongoing model development of the last decades. In this paper we evaluated the model's



performance with regard to a realistic representation of carbon and heat fluxes as well as ocean tracers in the UVic ESCM 2.10 in agreement with the available observational data and with current process understanding.

We find that the UVic ESCM 2.10 is capable of reproducing changes in historical temperature and carbon fluxes. There is a higher warming trend in the southern hemisphere south of 40S compare to observations, which causes a bias in SH sea ice trends. The simulated seasonal cycle of global mean temperature agrees well with the observed pattern, but has a lower amplitude. The air to sea fluxes of the UVic ESCM agree well with the observed pattern. The newly applied $CO_2$ forcing formulation has increased the model's climate sensitivity. Land carbon stocks concerning permafrost and vegetation carbon are within observational estimates, even though the spatial distribution of permafrost affected soil carbon and vegetation carbon densities show regional biases. The top of the atmosphere radiation balance of the UVic ESCM is well within the observed ranges, but the internal heat fluxes are slightly off. The simulated precipitation pattern shows good agreement with observations, but are regionally too spread out especially in the tropics and as expected do not reach the most extreme values. Terrestrial total precipitation and precipitation trends agree well with observations. Many ocean properties and tracers show good agreement with observations. This is mainly caused by a good representation of the general circulation. Although problems remain, mainly the too low Southern Ocean oxygen utilization and the salinity bias in the AAIW.

These model data deviations, especially for the ocean tracers have not yet been fully addressed (note the misfits have been reduced relative to the previous model version forced with new forcing), since we are already planning for the next update of the UVic ESCM, which will incorporate more comprehensive biogeochemical modules that will require re-tuning of the oceans biogeochemical parameters as well. Model developments that have not been incorporated in the model version described here, like carbon-nitrogen feedbacks on land (Wania et al., 2012), explicit representation of calcifiers in the ocean (Kvale et al., 2015), dynamic phosphorus cycle in the ocean (Niemeyer et al., 2017), and others, will in the following be implemented and tested within this new model version.

**Code availability** The model code is available on https://thredds.geomar.de/thredds/catalog/open_access/c565622a-9655-42bc-840c-c20e7dfd0861/catalog.html and will also be made available on the official UVic ESCM webpage (http://terra.seos.uvic.ca/model/), including all necessary documentation and data upon final publication of the manuscript.

**Data availability** The data show in the figures and used for this manuscript are made available on https://thredds.geomar.de/thredds/catalog/open_access/c565622a-9655-42bc-840c-c20e7dfd0861/catalog.html.

**Author contribution** All authors decided on the content of the new model version, DPK merged the model code, DPK, NW, KZ and NM compiled the CMIIP6 forcing input, NM ran the model and tuned it in consultation with ME, AM, KM, AS, HDM and KZ, NM wrote the manuscript with contributions from all authors.

**Competing Interest** The authors declare that they have no conflict of interest.

**Acknowledgments** The authors would like to acknowledge Claude-Michel Nzotungicimpaye for his helpful discussions about soil hydrology and carbon context, Alex MacIsaac for his discussions about the energy moisture balance model, Karin Kvale, Christopher Somes, and Wolfgang Koeve for her help looking at ocean alkalinity, and Heiner Dietze for his help with setting up the Fortran compiler. NM is grateful for support from the Natural Sciences and Engineering Research Council of Canada Discovery Grant (NSERC) grant awarded to K. Zickfeld, and the Helmholtz Initiative for Climate Adaptation and Mitigation (HI-CAM). KJM is thankful for funding by the Australian Research Council (DP180100048 and DP180102357). AHMD is grateful for support from Natural Sciences and Engineering Research Council of Canada Discovery Grant program.



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
