# Peer review of "Evaluation of the University of Victoria Earth System Climate Model version 2.10 (UVic ESCM 2.10)"

_Geoscientific Model Development, 2019_

## Referee Comment (RC1) · Anonymous Referee #1 · 15 Mar 2020

Review of Mengis et al. gmd-2019-373

This seems like a really straightforward behavior. The descriptions are excellent, and I'm quite impressed with the results of the model. I only have a few minor comments.

Abstract: Your abstract is pretty short and not all that specific. You have room to go into more details about some major developments or some details about how well the new version of the model performs.

Page 1, Lines 5-6: I think it would be helpful to be more specific here. More specifically, it is part of CMIP6, but the EMIC intercomparison – please say that.

Page 4, line 33: Could you include a definition or short description of cryoturbation?

Page 5, line 16: Change citep to citet.

Page 6, line 7: Please use proper scientific notation – this is a bit confusing as written.

Page 6, line 15: Some grammar issues

Page 8, line 8: Can you go into a few more details as to why this might be problematic?

Page 8, lines 26-27: It might be outside the EMIC range but is well within the ESM range. Is that coincidental or because the code modifications have resulted in the ability to capture more complex behavior?

―――――――――――――――

---

## Referee Comment (RC2) · Anonymous Referee #2 · 24 Mar 2020

**General comments**

This paper provides an overview of the latest update to the long-used UVic Earth System Model of Intermediate Complexity (version 2.10). The UVic model represents an impressive effort on behalf of many scientists in the Canadian climate modelling community and beyond. The previous version 2.9 has been used for many years in carbon budget assessments and modelling long-term climate change, applications for which full-complexity Earth system models are too computationally expensive. Particularly welcome is the inclusion of a permafrost module.

[Figure]

This update provides a valuable addition and extension to the UVic model, and it should be used extensively in the forthcoming IPCC assessments. Moreover, with a global focus on carbon budgets and net zero emissions, ESMs and EMICs that represent carbon cycle processes are even more valuable than previously. It should be published following the detailed minor revisions.

**Specific comments**

More should be done to convince the reader that in the era of increasing computing power, UVic is still a valuable model. Some indication of model runtime and the benefits of running UVic 2.10 versus a full-complexity ESM would be useful. What experiments can be done with UVic that ESMs would struggle with? Can you run perturbed parameter or perturbed physics ensembles (also leads into my next point)? On the other side of the coin, much emphasis is now being placed on simple climate models like MAGICC and FAIR, over which UVic has the advantage of fully representative physics, at least for the ocean and land surface.

The paper, to an extent, describes the tuning process for CMIP6. It is unclear which components of the model are hardwired and which are able to be changed according to the user's wishes. For example, can the aerosol forcing efficiency of sulphate optical depth to forcing be altered by the user? By default, the 1850-2018 aerosol ERF is around $-1.43$ W m$^{-2}$ in UVic 2.10. This is in fact stronger than all 12 CMIP6 ESMs and GCMs which have evaluated their aerosol forcing under CMIP6 emissions (Smith et al., 2020). Previous versions of UVic showed that metrics such as committed warming and climate sensitivity depend very strongly on the present-day aerosol forcing (Matthews and Zickfeld, 2012).

Related to this, on page 5, line 12: "All data used in the creation of this dataset can be accessed from Input4Mip on the Earth System Grid Federation (ESGF) unless otherwise specified." This slightly implies that we are constrained to use the CMIP6 tuning and/or forcing timeseries. Is this the case, or is this just for the tuning setup described in this paper?

Subheadings (i), (ii), (iii) in section 2.1 could be third level headings? (2.1.1, 2.1.2, 2.1.3)

Page 6, line 17: The AR5 aerosol effective radiative forcing range is $-1.9$ to $-0.1$ W m$^{-2}$. The appropriate references are Boucher et al., 2013 and Myhre et al., 2013, not the cited Ciais and Sabine.

Page 8, line 8: "might still be problematic": in what way? Could this affect long-term projections of near-surface air temperature?

Page 8, line 14: Temperature performance is evaluated with respect to the Global Warming Index (GWI), which seeks to isolate the anthropogenic component of historical warming from the natural component/total observed warming. But isn't natural forcing included in UVic? Section 2.1 suggests it is, and we can see the signatures of Krakatoa, Agung and Pinatubo in the temperature timeseries of figure 1a. On closer inspection, the curve marked as GWI seems to have a lot of internal variability, when it should be smooth like the yellow curve at http://globalwarmingindex.org/. Is this definitely GWI?

More pressingly, GWI is a mean of four datasets which are near-surface air temperature over land blended with sea-surface temperatures, of which three of the four have coverage bias by excluding regions where there are no observations such as the rapidly warming Arctic. If you are comparing UVic global near-surface air temperature over land and sea with no coverage masking (as is typically done when analysing temperature change in climate models) against GWI or any observational dataset without adjustment, you are likely underpredicting the present-day warming. See Richardson et al. (2016).

Page 8, line 26: It could be worth remarking on the interesting non-linearity between $2\times$ and $4\times CO_2$ ECS and possible mechanisms. Are these true long-term equilibrium values — presumably the model can be run long enough to determine this — or from a 150-year abrupt forcing (Gregory) regression?

Figure 2: Would it be better to plot summer minus winter and compare both hemispheres on one scale? A third plot showing the differences between UVic and obs would be nice too.

Supplementary figure S5: I'm not completely sure it's appropriate to copy the diagram from Wild et al., 2013 in fig. S5 unless you have permission from the publisher, but maybe you have obtained this.

**Technical corrections**

- line 9: "reproducing well changes" reads a little awkwardly, how about "reproducing changes in historical temperature and carbon fluxes well,"?

- page 3, line 3: Special Report on Global Warming of 1.5°C

- page 3, line 13: merge → merging

- page 3, line 25: period after (Weaver et al., 2001) rather than comma would improve flow in my opinion

- page 5, line 15: calculated → calculates

- page 5, line 16: Etminan reference as author (year)

- page 5, line 28: $\beta$ is better described as a forcing efficiency. $NO_x$ (NO and $NO_2$) are nitrogen oxides which are emitted in the gas rather than aerosol phase. As

you correctly allude, $NO_x$ are implicated in the formation of nitrate aerosols, as well as being an ozone precursor.

- page 6, line 2: Input4Mip → input4MIPs (also a few other places)

- page 6, eq. 4: $Ci → C_i$

- page 6, line 7: $E^{-5}$ and $E^{-3}$ would be better written e.g. $\times 10^{-5}$.

- page 7, line 2: the IMAGE model

- page 7, line 8: close bracket has no corresponding open bracket

- page 8, line 20: CO2 → $CO_2$ (also a few other places)

- page 8, line 21: remove "with"

- page 9, lines 1-2: written a little awkwardly. Generally, it would be good to be consistent with units throughout the paper: use either K or °C consistently.

- page 11, line 8: than in → compared to

- page 12, table 3: solar down range should be 340 not 240?

- page 12, line 7: remove "with"

- page 12, line 18: at → is at

- page 12, line 20: simulates → simulated

- page 12, line 24: consistency with citing IPCC chapters

- page 18, figure 9 caption: Taylor 2001 is not in the list of references.

[Figure]

- page 22, line 15: Capitalise Coupled Model Intercomparison Project, although will UVic actually be used in CMIP6? I would assume ZECMIP and C4MIP? If so I would also suggest highlighting this in the abstract too.

- page 23, line 10: "are slightly off": can this be written more scienficially?

**References**

Boucher, O., D. Randall, P. Artaxo, C. Bretherton, G. Feingold, P. Forster, V.-M. Kerminen, Y. Kondo, H. Liao, U. Lohmann, P. Rasch, S.K. Satheesh, S. Sherwood, B. Stevens and X.Y. Zhang, 2013: Clouds and Aerosols. In: Climate Change 2013: The Physical Science Basis. Contribution of Working Group I to the Fifth Assessment Report of the Intergovernmental Panel on Climate Change [Stocker, T.F., D. Qin, G.-K. Plattner, M. Tignor, S.K. Allen, J. Boschung, A. Nauels, Y. Xia, V. Bex and P.M. Midgley (eds.)]. Cambridge University Press, Cambridge, United Kingdom and New York, NY, USA

Myhre, G., D. Shindell, F.-M. Bréon, W. Collins, J. Fuglestvedt, J. Huang, D. Koch, J.-F. Lamarque, D. Lee, B. Mendoza, T. Nakajima, A. Robock, G. Stephens, T. Takemura and H. Zhang, 2013: Anthropogenic and Natural Radiative Forcing. In: Climate Change 2013: The Physical Science Basis. Contribution of Working Group I to the Fifth Assessment Report of the Intergovernmental Panel on Climate Change [Stocker, T.F., D. Qin, G.-K. Plattner, M. Tignor, S.K. Allen, J. Boschung, A. Nauels, Y. Xia, V. Bex and P.M. Midgley (eds.)]. Cambridge University Press, Cambridge, United Kingdom and New York, NY, USA.

Smith, C. J., Kramer, R. J., Myhre, G., Alterskjær, K., Collins, W., Sima, A., Boucher, O., Dufresne, J.-L., Nabat, P., Michou, M., Yukimoto, S., Cole, J., Paynter, D., Shiogama, H., O'Connor, F. M., Robertson, E., Wiltshire, A., Andrews, T., Hannay, C., Miller, R., Nazarenko, L., Kirkevåg, A., Olivié, D., Fiedler, S., Pincus, R., and Forster, P. M.: Effective radiative forcing and adjustments in CMIP6 models, Atmos. Chem. Phys. Discuss., https://doi.org/10.5194/acp-2019-1212, in review, 2020.

Matthews, H., Zickfeld, K. Climate response to zeroed emissions of greenhouse gases and aerosols. Nature Clim Change 2, 338–341 (2012). https://doi.org/10.1038/nclimate1424

Richardson, M., Cowtan, K., Hawkins, E. and Stolpe, M.B. Reconciled climate response estimates from climate models and the energy budget of Earth. Nature Clim Change 6, 931–935 (2016). https://doi.org/10.1038/nclimate3066

---

## Author Comment (AC1) · 8 Jun 2020

We would like to thank the reviewers for the positive evaluation of this effort and the very constructive comments which further improved this manuscript. We followed all the reviewer suggestions (for details see below, or attachment rebuttal_200605.pdf). In addition to the reviewers suggestions, we decided to add a schematic outlining the different components included in the University of Victoria Earth system Climate model version 2.10 (Fig 1).

We hope that the submitted material is a valuable contribution to Geoscientific Model Development.

Best regards, Nadine Mengis

Anonymous Referee 1

Review of Mengis et al. gmd-2019-373

This seems like a really straightforward behavior. The descriptions are excellent, and I'm quite impressed with the results of the model. I only have a few minor comments.

**Thank you for this positive evaluation of our manuscript.**

Abstract: Your abstract is pretty short and not all that specific. You have room to go into more details about some major developments or some details about how well the new version of the model performs.

**Thank you, that is a good point. We added some sentences accordingly.**
**"The main additions to the base model are: i) an improved biogeochemistry module for the ocean, ii) a vertically resolved soil model including dynamic hydrology and soil carbon processes, and iii) a representation of permafrost carbon."**
**"For the moment, the main biases that remain are a vegetation carbon density that is too high in the tropics, a higher than observed change in the ocean heat content, and a oxygen utilization in the Southern Ocean that is too low. All of these biases will be addressed in the next updates to the model."**

Page 1, Lines 5-6: I think it would be helpful to be more specific here. More specifically, it is part of CMIP6, but the EMIC intercomparison – please say that.

**We added some more information:**
**"The new version 2.10 of the University of Victoria Earth System Climate Model (UVic ESCM) presented here, will be part of the 6th phase of the coupled model intercomparison project (CMIP6). More precisely it will be used in the intercomparison of Earth system models of intermediate complexity (EMICs), such as the C4MIP, the Carbon Dioxide Removal and Zero Emissions Commitment model intercomparison projects (CDR-MIP and ZECMIP, respectively)."**

Page 4, line 33: Could you include a definition or short description of cryoturbation?

**Sure, no problem:**
**"A representation of permafrost carbon has also been added to the model. Permafrost carbon is prognostically generated within the model using a diffusion-based scheme meant to approximate the process of cryoturbation (MacDougall Knutti 2016). That is, a freeze-thaw generated mechanical mixing process that causes subduction of organic carbon rich soils from the surface into deeper soil layers in permafrost affected soils."**

Page 5, line 16: Change citep to citet.

**Thank you. We changed this.**

Page 6, line 7: Please use proper scientific notation – this is a bit confusing as written.

**Thank you. We changed this.**

Page 6, line 15: Some grammar issues

**Thank you for pointing this out. We changed it to:**
**"The resulting AOD caused a forcing that was too strong in the historical period. Therefore, an option was implemented into the UVic ESCM, which allows the user to scale the aerosol forcing from AOD data to fit it to current values."**

Page 8, line 8: Can you go into a few more details as to why this might be problematic?

**We added a more detailed discussion on this issue:**
**"An overestimation in the change in ocean heat content anomaly would for example result in a similar overestimation of thermosteric sea level change. Another possible impact of the overestimated ocean heat uptake can be the estimates of the Zero Emissions Commitment, which is directly linked to the state of thermal equilibration of the Earth system (MacDougall et al., 2020). So the fact that EMICs in general, but the UVic ESCM 2.10 in particular here overestimates the OHC anomaly trend has to be kept in mind if the model were used for experiments concerning this metric."**

Page 8, lines 26-27: It might be outside the EMIC range but is well within the ESM range. Is that coincidental or because the code modifications have resulted in the ability to capture more complex behavior?

The code has been changed only with respect to the CO2 forcing formulation to accommodate the new findings from Etminan et al., 2016. Otherwise the model is still using its rather simplistic atmosphere representation. Note that the numbers have changed due to a correction of the sediment fluxes. They now are well within the range reported by Eby et al., 2013.

"For all transient and diagnostic simulations, the weathering flux was then set constant to the value at the end of the spin-up of 8703 kg C s-1."

Anonymous Referee 2

General comments
This paper provides an overview of the latest update to the long-used UVic Earth
System Model of Intermediate Complexity (version 2.10). The UVic model represents
an impressive effort on behalf of many scientists in the Canadian climate modelling
community and beyond. The previous version 2.9 has been used for many years in
carbon budget assessments and modelling long-term climate change, applications
for which full-complexity Earth system models are too computationally expensive.
Particularly welcome is the inclusion of a permafrost module.
This update provides a valuable addition and extension to the UVic model, and it
should be used extensively in the forthcoming IPCC assessments. Moreover, with
a global focus on carbon budgets and net zero emissions, ESMs and EMICs that
represent carbon cycle processes are even more valuable than previously. It should
be published following the detailed minor revisions.

**We would like to thank the reviewer for this positive evaluation of our effort.**

Specific comments
More should be done to convince the reader that in the era of increasing computing
power, UVic is still a valuable model. Some indication of model runtime and the
benefits of running UVic 2.10 versus a full-complexity ESM would be useful. What
experiments can be done with UVic that ESMs would struggle with? Can you run
perturbed parameter or perturbed physics ensembles (also leads into my next point)?
On the other side of the coin, much emphasis is now being placed on simple climate
models like MAGICC and FAIR, over which UVic has the advantage of fully represen-
tative physics, at least for the ocean and land surface.

**Thank you for this suggestion. We added a paragraph on the advantages of using EMICs and the UVic in particular.**
**"As an Earth system model of intermediate complexity, the UVic ESCM has a comparably low computational cost (4.6-11.5hours per 100years on a simple desktop computer, depending on the computational power of the machine), while still providing a comprehensive carbon cycle model with a full represented ocean physics. It is therefore a well-suited tool to for example perform large perturbed parameter ensembles to constrain process level uncertainties (e.g., MacDougallKnutti, 2016; Mengis et al., 2018). Such experiments are not yet feasibly in a state-of-the-art Earth system model (ESM). Thanks to its representation of many important components of the carbon cycle and the physical climate and its ability to simulate dynamic interactions between them, the UVic ESCM is a more comprehensive tool for uncertainty assessment compared to the simple climate models such as MAGICC."**

The paper, to an extent, describes the tuning process for CMIP6. It is unclear which components of the model are hardwired and which are able to be changed according to the user's wishes. For example, can the aerosol forcing efficiency of sulphate optical depth to forcing be altered by the user?

**We tried to make this clearer throughout the text. For example we reformulated the aerosol option:**
**"Therefore, an option was implemented into the UVic ESCM, which allows the user to scale the aerosol forcing from AOD data to fit it to current values."**

By default, the 1850-2018 aerosol ERF is around $-1.43$ W m$-2$ in UVic 2.10. This

is in fact stronger than all 12 CMIP6 ESMs and GCMs which have evaluated their aerosol forcing under CMIP6 emissions (Smith et al., 2020). Previous versions of UVic showed that metrics such as committed warming and climate sensitivity depend very strongly on the present-day aerosol forcing (Matthews and Zickfeld, 2012).

**That is correct, the default forcing is too strong, which is why we scaled it down: "For transient simulations, the scaling factor was set to 0.7, which gives a globally average forcing of -1.04 Wm-2 in 2014, consistent with the IPCC AR5 range estimate of between -2.3 and 0.2 Wm-2 (Ciais and Sabine, 2013) and the newest updates of this forcing of -1.04 $\pm$ 0.23 Wm-2 from Smith et al. (2020)."**
**Now, this option allows the user to really easily apply sensitivity analysis of this forcing component, and its influence on e.g. the ZEC.**

Related to this, on page 5, line 12: "All data used in the creation of this dataset can be accessed from Input4Mip on the Earth System Grid Federation (ESGF) unless otherwise specified." This slightly implies that we are constrained to use the CMIP6 tuning and/or forcing timeseries. Is this the case, or is this just for the tuning setup described in this paper?

**Thank you for this comment. This section is merely meant to point to the data source for the CMIP6 forcing. The model as described here has been tuned to be spun-up and to perform well using CMIP6 data, spin-up protocol and forcing protocol. As there have been substantial updates especially concerning solar data this was a non-trivial task. If the model would be applied in another context, for example in a paleo study it is advised to evaluate the models behaviour to reproduce the respective data. But it does not mean it is limited to be used in the CMIP6 context only. We added a sentence on this at the end of this section: "Even though the model has been fine tuned to reproduce the recent observa-**

tional period while following the CMIP6 forcing data and protocols, the model is not limited to CMIP6 context applications."

Subheadings (i), (ii), (iii) in section 2.1 could be third level headings? (2.1.1, 2.1.2, 2.1.3)
**We changed this.**

Page 6, line 17: The AR5 aerosol effective radiative forcing range is $-1.9$ to $-0.1$ W m$-2$. The appropriate references are Boucher et al., 2013 and Myhre et al., 2013, not the cited Ciais and Sabine.

**Thank you for providing the correct citations!**

Page 8, line 8: "might still be problematic": in what way? Could this affect long-term projections of near-surface air temperature?

**In the long term this is a possibility. A positive bias in the surface ocean heat capacity change points to the fact that the ocean takes up more heat in the simulations compared with what is observed. This in turn might then impact the thermal equilibration state at e.g. the point of zero emissions and thereby the estimate for the zero emissions commitment. We added a sentence here to discuss possible pitfalls.**
**"This seems to be a general feature of EMICs (Eby et al., 2013), but might still be problematic. An overestimation in the change in ocean heat content anomaly would for example result in a similar overestimation of thermosteric sea level change. Another possible impact of the overestimated ocean heat uptake can be the estimates of the Zero Emissions Commitment, which is directly linked to**

**the state of thermal equilibration of the Earth system. So the fact that EMICs in general, but the UVic ESCM 2.10 in particular here overestimates the OHC anomaly trend has to be kept in mind if the model were used for experiments concerning this metric."**

Page 8, line 14: Temperature performance is evaluated with respect to the Global Warming Index (GWI), which seeks to isolate the anthropogenic component of historical warming from the natural component/total observed warming. But isn't natural forcing included in UVic? Section 2.1 suggests it is, and we can see the signatures of Krakatoa, Agung and Pinatubo in the temperature timeseries of figure 1a. On closer inspection, the curve marked as GWI seems to have a lot of internal variability, when it should be smooth like the yellow curve at http://globalwarmingindex.org/. Is this definitely GWI?
More pressingly, GWI is a mean of four datasets which are near-surface air temperature over land blended with sea-surface temperatures, of which three of the four have coverage bias by excluding regions where there are no observations such as the rapidly warming Arctic. If you are comparing UVic global near-surface air temperature over land and sea with no coverage masking (as is typically done when analysing temperature change in climate models) against GWI or any observational dataset without adjustment, you are likely under-predicting the present-day warming. See Richardson et al. (2016).

**Thank you for clarifying this! We have previously been using the average of all four datasets, which as you pointed out would likely introduce a coverage bias when compared to the GSAT as calculated in our model. What we show now is HadCRUT4-CW filled in dataset, in comparison with the global mean surface air temperature as simulated by our model.**

Page 8, line 26: It could be worth remarking on the interesting non-linearity between 2× and 4×CO2 ECS and possible mechanisms. Are these true long-term equilibrium values — presumably the model can be run long enough to determine this — or from a 150-year abrupt forcing (Gregory) regression?

**Thank you for this comment. The reviewer is right to assume that these are equilibrium values for simulations that were simulated for 1000 years. The previously found non-linearity has decreased substantially in our new simulations. It turned out that this was due to multiple reasons, one being the sea ice albedo feedback that the 2xCO2 simulation had experience, and another being the terrestrial weathering flux that had not been set correctly for all transient simulations. An error we have now corrected. Because of this, we did not comment further on it in this manuscript. We changed the sentence to:**
**"In the same way, there is a good agreement with the EMIC multi-model mean and the diagnosed values for the Equilibrium Climate Sensitivity for a 2x and 4x increase in atmospheric CO2 concentrations, with temperature increases of 3.39 °C and 6.47 °C, respectively."**

Figure 2: Would it be better to plot summer minus winter and compare both hemispheres on one scale? A third plot showing the differences between UVic and obs would be nice too.

**We changed the plot accordingly and added the latitudinal means. We hope that this is sufficient for the comparison of the UVic and the observational data set.**

Supplementary figure S5: I'm not completely sure it's appropriate to copy the diagram from Wild et al., 2013 in fig. S5 unless you have permission from the publisher, but

maybe you have obtained this.

**To be on the safe side we did our own version of the radiation balance, motivated by Wild et al., 2013. Thank you for pointing this out!**

**Thank you for the very helpful and detail-oriented technical corrections. We implemented all of them in the revised version.**

Technical corrections

line 9: "reproducing well changes" reads a little awkwardly, how about "reproducing changes in historical temperature and carbon fluxes well,"?
**Done.**
page 3, line 3: Special Report on Global Warming of 1.5◦C
**Done.**
page 3, line 13: merge → merging
**Done.**
page 3, line 25: period after (Weaver et al., 2001) rather than comma would improve flow in my opinion
**Done.**
page 5, line 15: calculated → calculates
**Done.**
page 5, line 16: Etminan reference as author (year)
**Done.**
page 5, line 28: $\beta$ is better described as a forcing efficiency. NOx (NO and NO2)

are nitrogen oxides which are emitted in the gas rather than aerosol phase. As you correctly allude, NOx are implicated in the formation of nitrate aerosols, as well as being an ozone precursor.

**We changed this formulation.**

page 6, line 2: Input4Mip → input4MIPs (also a few other places)

**Done.**

page6,eq. 4: Ci→Ci

**Done.**

page 6, line 7: E−5 and E−3 would be better written e.g. ×10−5.

**Done.**

page 7, line 2: the IMAGE model

**Done.**

page 7, line 8: close bracket has no corresponding open bracket

**Done.**

page 8, line 20: CO2 → CO2 (also a few other places)

**We changed this. Note that we left the non-subscripted version anywhere where we refer to the 2xCO2 and 4xCO2 experiments.**

page 8, line 21: remove "with"

**Done.**

page 9, lines 1-2: written a little awkwardly. Generally, it would be good to be consistent with units throughout the paper: use either K or ◦C consistently.

**Done.**

page 11, line 8: than in → compared to

**Done.**

page 12, table 3: solar down range should be 340 not 240?

**Wow. Thank you for catching this one!**

page 12, line 7: remove "with"

**Done.**

page13,line18: at→isat

**Done.**
page 13, line 20: simulates → simulated
**Done.**
page 13, line 24: consistency with citing IPCC chapters
**Done.**
page 18, figure 9 caption: Taylor 2001 is not in the list of references.
**Thank you!! We added this.**
page 22, line 15: Capitalise Coupled Model Intercomparison Project, although will
UVic actually be used in CMIP6? I would assume ZECMIP and C4MIP? If so I would
also suggest highlighting this in the abstract too.
**Thank you for the suggestion. We added this to the abstract.**
**"The new version 2.10 of the University of Victoria Earth System Climate Model
(UVic ESCM) presented here, will be part of the 6th phase of the coupled
model intercomparison project (CMIP6). More precisely it will be used in the
intercomparison of Earth system models of intermediate complexity (EMICs),
like the C4MIP, the Carbon Dioxide Removal and Zero Emissions Commitment
model intercomparison projects (CDR-MIP and ZECMIP, respectively)."**
page 23, line 10: "are slightly off": can this be written more scientifically?
**Done.**

Please also note the supplement to this comment:
https://www.geosci-model-dev-discuss.net/gmd-2019-373/gmd-2019-373-AC1-
supplement.pdf
* * *
**UVic ESCM**

Atmospheric Model

Sea Ice
Model

Vegetation
Model

Permafrost
Model

Ocean General
Circulation Model

Land Surface
Model

N, P, O
Cycling

Organic
Carbon Cycle

Inorganic
Carbon Cycle

Marine
Sediment Model

Artwork: Rita Erven, GEOMAR

Common
horizontal resolution
3.6° longitude x 1.8° latitude

Energy    Water    Carbon

**Fig. 1.** Figure 1: Schematic of the University of Victoria Earth System Climate Model version 2.10 (UVic ESCM 2.10).

**Supplement:**

Review of Mengis et al. gmd-2019-373

This seems like a really straightforward behavior. The descriptions are excellent, and I'm quite impressed with the results of the model. I only have a few minor comments.

Thank you for this positive evaluation of our manuscript.

Abstract: Your abstract is pretty short and not all that specific. You have room to go into more details about some major developments or some details about how well the new version of the model performs.

Thank you. That is a good point. We added some sentences accordingly.

"The main additions to the base model are: i) an improved biogeochemistry module for the ocean, ii) a vertically resolved soil model including dynamic hydrology and soil carbon processes, and iii) a representation of permafrost carbon."

"For the moment, the main biases that remain are a vegetation carbon density that is too high in the tropics, a higher than observed change in the ocean heat content, and a oxygen utilization in the Southern Ocean that is too low. All of these biases will be addressed in the next updates to the model."

Page 1, Lines 5-6: I think it would be helpful to be more specific here. More specifically, it is part of CMIP6, but the EMIC intercomparison – please say that.

We added some more information:

"The new version 2.10 of the University of Victoria Earth System Climate Model (UVic ESCM) presented here, will be part of the 6[th] phase of the coupled model intercomparison project (CMIP6). More precisely it will be used in the intercomparison of Earth system models of intermediate complexity (EMICs), such as the C4MIP, the Carbon Dioxide Removal and Zero Emissions Commitment model intercomparison projects (CDR-MIP and ZECMIP, respectively)."

Page 4, line 33: Could you include a definition or short description of cryoturbation?

Sure, no problem:

"A representation of permafrost carbon has also been added to the model. Permafrost carbon is prognostically generated within the model using a diffusion-based scheme meant to approximate the process of cryoturbation (MacDougall & Knutti 2016). That is, a freeze-thaw generated mechanical mixing process that causes subduction of organic carbon rich soils from the surface into deeper soil layers in permafrost affected soils."

Page 5, line 16: Change citep to citet.

Thank you. We changed this.

Page 6, line 7: Please use proper scientific notation – this is a bit confusing as written.

Thank you. We changed this.

Page 6, line 15: Some grammar issues

Thank you for pointing this out. We changed it to:

"The resulting AOD caused a forcing that was too strong in the historical period. Therefore, an option was implemented into the UVic ESCM, which allows the user to scale the aerosol forcing from AOD data to fit it to current values."

Page 8, line 8: Can you go into a few more details as to why this might be problematic?

We added a more detailed discussion on this issue:

"An overestimation in the change in ocean heat content anomaly would for example result in a similar overestimation of thermosteric sea level change. Another possible impact of the overestimated ocean heat uptake can be the estimates of the Zero Emissions Commitment, which is directly linked to the state of thermal equilibration of the Earth system \citep{MacDougallEtAl2020}. So the fact that EMICs in general, but the UVic ESCM 2.10 in particular here overestimates the OHC anomaly trend has to be kept in mind if the model were used for experiments concerning this metric."

Page 8, lines 26-27: It might be outside the EMIC range but is well within the ESM range. Is that coincidental or because the code modifications have resulted in the ability to capture more complex behavior?

The code has been changed only with respect to the CO2 forcing formulation to accommodate the new findings from Etminan et al., 2016. Otherwise the model is still using its rather simplistic atmosphere representation. Note that the numbers have changed due to a correction of the sediment fluxes. They now are well within the range reported by Eby et al., 2013.

"For all transient and diagnostic simulations, the weathering flux was then set constant to the value at the end of the spin-up of 8703 kg C s$^{-1}$."

**Anonymous Referee #2**

General comments

This paper provides an overview of the latest update to the long-used UVic Earth System Model of Intermediate Complexity (version 2.10). The UVic model represents an impressive effort on behalf of many scientists in the Canadian climate modelling community and beyond. The previous version 2.9 has been used for many years in carbon budget assessments and modelling long-term climate change, applications for which full-complexity Earth system models are too computationally expensive. Particularly welcome is the inclusion of a permafrost module.

This update provides a valuable addition and extension to the UVic model, and it should be used extensively in the forthcoming IPCC assessments. Moreover, with a global focus on carbon budgets and net zero emissions, ESMs and EMICs that represent carbon cycle processes are even more valuable than previously. It should be published following the detailed minor revisions.

We would like to thank the reviewer for this positive evaluation of our effort.

Specific comments

More should be done to convince the reader that in the era of increasing computing power, UVic is still a valuable model. Some indication of model runtime and the benefits of running UVic 2.10 versus a full-complexity ESM would be useful. What experiments can be done with UVic that ESMs would struggle with? Can you run perturbed parameter or perturbed physics ensembles (also leads into my next point)? On the other side of the coin, much emphasis is now being placed on simple climate models like MAGICC and FAIR, over which UVic has the advantage of fully representative physics, at least for the ocean and land surface.

Thank you for this suggestion. We added a paragraph on the advantages of using EMICs and the UVic in particular.

"As an Earth system model of intermediate complexity, the UVic ESCM has a comparably low computational cost (4.6-11.5hours per 100years on a simple desktop computer, depending on the computational power of the machine), while still providing a comprehensive carbon cycle model with a full represented ocean physics. It is therefore a well-suited tool to for example perform large perturbed parameter ensembles to constrain process level uncertainties (e.g., MacDougallKnutti, 2016; Mengis et al., 2018). Such experiments are not yet feasibly in a state-of-the-art Earth system model (ESM). Thanks to its representation of many important components of the carbon cycle and the physical climate and its ability to simulate dynamic interactions between them, the UVic ESCM is a more comprehensive tool for uncertainty assessment compared to the simple climate models such as MAGICC."

The paper, to an extent, describes the tuning process for CMIP6. It is unclear which components of the model are hardwired and which are able to be changed according to the user's wishes. For example, can the aerosol forcing efficiency of sulphate optical depth to forcing be altered by the user?

We tried to make this clearer throughout the text. For example we reformulated the aerosol option:

"Therefore, an option was implemented into the UVic ESCM, which allows the user to scale the aerosol forcing from AOD data to fit it to current values."

By default, the 1850-2018 aerosol ERF is around −1.43 W m−2 in UVic 2.10. This is in fact stronger than all 12 CMIP6 ESMs and GCMs which have evaluated their aerosol forcing under CMIP6 emissions (Smith et al., 2020). Previous versions of UVic showed that metrics such as committed warming and climate sensitivity depend very strongly on the present-day aerosol forcing (Matthews and Zickfeld, 2012).

That is correct, the default forcing is too strong, which is why we scaled it down. ꞏSEPꞏSEPꞏ "For transient simulations, the scaling factor was set to 0.7, which gives a globally average forcing of -1.04 Wm⁻² in 2014, consistent with the IPCC AR5 range estimate of between -2.3 and 0.2 Wm⁻² (Ciais and Sabine, 2013) and the newest updates of this forcing of -1.04 ± 0.23 Wm⁻² from Smith et al. (2020)."

Now, this option allows the user to really easily apply sensitivity analysis of this forcing component, and its influence on e.g. the ZEC.

Related to this, on page 5, line 12: "All data used in the creation of this dataset can be accessed from Input4Mip on the Earth System Grid Federation (ESGF) unless otherwise specified." This slightly implies that we are constrained to use the CMIP6 tuning and/or forcing timeseries. Is this the case, or is this just for the tuning setup described in this paper?

Thank you for this comment. This section is merely meant to point to the data source for the CMIP6 forcing. The model as described here has been tuned to be spun-up and to perform well using CMIP6 data, spin-up protocol and forcing protocol. As there have been substantial updates especially concerning solar data this was a non-trivial task. If the model would be applied in another context, for example in a paleo study it is advised to evaluate the models behaviour to reproduce the respective data. But it does not mean it is limited to be used in the CMIP6 context only. We added a sentence on this at the end of this section:

"Even though the model has been fine tuned to reproduce the recent observational period while following the CMIP6 forcing data and protocols, the model is not limited to CMIP6 context applications."

Subheadings (i), (ii), (iii) in section 2.1 could be third level headings? (2.1.1, 2.1.2, 2.1.3)

We changed this.

Page 6, line 17: The AR5 aerosol effective radiative forcing range is −1.9 to −0.1 W m−2. The appropriate references are Boucher et al., 2013 and Myhre et al., 2013, not the cited Ciais and Sabine.

Thank you for providing the correct citations!

Page 8, line 8: "might still be problematic": in what way? Could this affect long-term projections of near-surface air temperature?

In the long term this is a possibility. A positive bias in the surface ocean heat capacity change points to the fact that the ocean takes up more heat in the simulations compared with what is observed. This in turn might then impact the thermal equilibration state at e.g. the point of zero emissions and thereby the estimate for the zero emissions commitment. We added a sentence here to discuss possible pitfalls.

"This seems to be a general feature of EMICs (Eby et al., 2013), but might still be problematic. An overestimation in the change in ocean heat content anomaly would for example result in a similar overestimation of thermosteric sea level change. Another possible impact of the overestimated ocean heat uptake can be the estimates of the Zero Emissions Commitment, which is directly linked to the state of

thermal equilibration of the Earth system. So the fact that EMICs in general, but the UVic ESCM 2.10 in particular here overestimates the OHC anomaly trend has to be kept in mind if the model were used for experiments concerning this metric."

Page 8, line 14: Temperature performance is evaluated with respect to the Global Warming Index (GWI), which seeks to isolate the anthropogenic component of historical warming from the natural component/total observed warming. But isn't natural forcing included in UVic? Section 2.1 suggests it is, and we can see the signatures of Krakatoa, Agung and Pinatubo in the temperature timeseries of figure 1a. On closer inspection, the curve marked as GWI seems to have a lot of internal variability, when it should be smooth like the yellow curve at http://globalwarmingindex.org/. Is this definitely GWI?

More pressingly, GWI is a mean of four datasets which are near-surface air temperature over land blended with sea-surface temperatures, of which three of the four have coverage bias by excluding regions where there are no observations such as the rapidly warming Arctic. If you are comparing UVic global near-surface air temperature over land and sea with no coverage masking (as is typically done when analysing temperature change in climate models) against GWI or any observational dataset without adjustment, you are likely under-predicting the present-day warming. See Richardson et al. (2016).

Thank you for clarifying this! We have previously been using the average of all four datasets, which as you pointed out would likely introduce a coverage bias when compared to the GSAT as calculated in our model. What we show now is HadCRUT4-CW filled in dataset, in comparison with the global mean surface air temperature as simulated by our model.

Page 8, line 26: It could be worth remarking on the interesting non-linearity between 2× and 4×CO2 ECS and possible mechanisms. Are these true long-term equilibrium values — presumably the model can be run long enough to determine this — or from a 150-year abrupt forcing (Gregory) regression?

Thank you for this comment. The reviewer is right to assume that these are equilibrium values for simulations that were simulated for 1000 years. The previously found non-linearity has decreased substantially in our new simulations. It turned out that this was due to multiple reasons, one being the sea ice albedo feedback that the 2xCO2 simulation had experience, and another being the terrestrial weathering flux that had not been set correctly for all transient simulations. An error we have now corrected. Because of this, we did not comment further on it in this manuscript. We changed the sentence to:

"In the same way, there is a good agreement with the EMIC multi-model mean and the diagnosed values for the Equilibrium Climate Sensitivity for a 2x and 4x increase in atmospheric $CO_2$ concentrations, with temperature increases of 3.39 °C and 6.47 °C, respectively."

Figure 2: Would it be better to plot summer minus winter and compare both hemispheres on one scale? A third plot showing the differences between UVic and obs would be nice too.

We changed the plot accordingly and added the latitudinal means. We hope that this is sufficient for the comparison of the UVic and the observational data set.

Supplementary figure S5: I'm not completely sure it's appropriate to copy the diagram from Wild et al., 2013 in fig. S5 unless you have permission from the publisher, but maybe you have obtained this.

To be on the safe side we did our own version of the radiation balance, motivated by Wild et al., 2013. Thank you for pointing this out!

Thank you for the very helpful and detail-oriented technical corrections. We implemented all of them in the revised version.

**Technical corrections**

line 9: "reproducing well changes" reads a little awkwardly, how about "reproduc-ing changes in historical temperature and carbon fluxes well,"?

Done.

page 3, line 3: Special Report on Global Warming of 1.5∘C

Done.

page 3, line 13: merge → merging

Done.

page 3, line 25: period after (Weaver et al., 2001) rather than comma would improve flow in my opinion

Done.

page 5, line 15: calculated → calculates

Done.

page 5, line 16: Etminan reference as author (year)

Done.

page 5, line 28: $\beta$ is better described as a forcing efficiency. NOx (NO and NO2) are nitrogen oxides which are emitted in the gas rather than aerosol phase. As you correctly allude, NOx are implicated in the formation of nitrate aerosols, as well as being an ozone precursor.

We changed this formulation.

page 6, line 2: Input4Mip → input4MIPs (also a few other places)

Done.

page6,eq. 4: Ci→$C_i$

Done.

page 6, line 7: E−5 and E−3 would be better written e.g. ×10−5.

Done.

page 7, line 2: the IMAGE model

Done.

page 7, line 8: close bracket has no corresponding open bracket

Done.

page 8, line 20: CO2 → $CO_2$ (also a few other places)

We changed this. Note that we left the non-subscripted version anywhere where we refer to the 2xCO2 and 4xCO2 experiments.

page 8, line 21: remove "with"

Done.

page 9, lines 1-2: written a little awkwardly. Generally, it would be good to be consistent with units throughout the paper: use either K or ∘C consistently.

Done.

page 11, line 8: than in → compared to

Done.

page 12, table 3: solar down range should be 340 not 240?

Wow. Thank you for catching this one!

page 12, line 7: remove "with"

Done.

page13,line18: at→isat

Done.

page 13, line 20: simulates → simulated Done.

page 13, line 24: consistency with citing IPCC chapters

Done.

page 18, figure 9 caption: Taylor 2001 is not in the list of references.

Thank you!! We added this.

page 22, line 15: Capitalise Coupled Model Intercomparison Project, although will UVic actually be used in CMIP6? I would assume ZECMIP and C4MIP? If so I would also suggest highlighting this in the abstract too.

Thank you for the suggestion. We added this to the abstract.

"The new version 2.10 of the University of Victoria Earth System Climate Model (UVic ESCM) presented here, will be part of the 6[th] phase of the coupled model intercomparison project (CMIP6). More precisely it will be used in the intercomparison of Earth system models of intermediate complexity (EMICs), like the C4MIP, the Carbon Dioxide Removal and Zero Emissions Commitment model intercomparison projects (CDR-MIP and ZECMIP, respectively)."

page 23, line 10: "are slightly off": can this be written more scientifically?

Done.

[revised manuscript text omitted]

---

## Author Response (AR2)

**Author's Response to the comments by the Topical Editor Decision: Publish subject to minor revisions (review by editor) (22 Jun 2020)**

Review of Mengis et al. gmd-2019-373

Dear Mr Phipps,

Thank you for pointing this out! It seems that the link I provided worked well for GEOMAR internal persons only. I clarified this with our IT department, which provided me with an updated link. When we tested this link again, it worked well. Please let me know in case of any difficulties accessing the data or code.

Best regards,

Nadine Mengis

---

## Author Response (AR3)

**Author's Response to the comments by the Topical Editor Decision: Publish subject to minor revisions (review by editor) (22 Jun 2020)**

Review of Mengis et al. gmd-2019-373

Dear Mr Phipps,

One of our students realised that we have been using $CO_2$ emissions for the emissions driven run, that are based on CMIP5 historical emissions (that are very close to the emissions as reported by the Global Carbon Budget 2018).
In the paper however, we report that we are using CMIP6 $CO_2$ emissions.
We have performed a simulation and the differences at least in terms of Global Mean Air temperature are very small. And we do believe that the results and the conclusion of the paper will not be impacted.

In the revised manuscript version we use our new simulations and edit the plots in order to stay consistent with the new CMIP6 forcing. We also added the student who performed the new $CO_2$ emissions driven simulation to the list of co-authors: Alexander J. MacIsaac.

Best regards,

Nadine Mengis